# TELL ME WHAT YOU DON'T KNOW: ENHANCING REFUSAL CAPABILITIES OF ROLE-PLAYING AGENTS VIA REPRESENTATION SPACE ANALYSIS AND EDITING

## ABSTRACT

Role-Playing Agents (RPAs) have shown remarkable performance in various applications, yet they often struggle to recognize and appropriately respond to hard queries that conflict with their role-play knowledge. To investigate RPAs' performance when faced with different types of conflicting requests, we develop an evaluation benchmark that includes contextual knowledge conflicting requests, parametric knowledge conflicting requests, and non-conflicting requests to assess RPAs' ability to identify conflicts and refuse to answer appropriately without over-refusing. Through extensive evaluation, we find that most RPAs behave significant performance gaps toward different conflict requests. To elucidate the reasons, we conduct an in-depth representation-level analysis of RPAs under various conflict scenarios. Our findings reveal the existence of **rejection regions** and **direct response regions** within the model's forwarding representation, and thus influence the RPA's final response behavior. Therefore, we introduce a lightweight representation editing approach that conveniently shifts conflicting requests to the rejection region, thereby enhancing the model's refusal accuracy. The experimental results validate the effectiveness of our editing method, improving RPAs' refusal ability of conflicting requests while maintaining their general role-playing capabilities.

## 1 INTRODUCTION

Role-Playing Agents(RPAs), ranging from non-player characters in video games(Wang et al., 2023a) to virtual assistants(Tseng et al., 2024) and interactive educational tools(Wei et al., 2024), are revolutionizing human-computer interaction(Chen et al., 2024b). The growing importance of RPAs in AI applications underscores the need to improve their performance. Previous work in the field of role-playing has primarily focused on enhancing the performance of RPAs through techniques such as prompt-based methods and fine-tuning(Wang et al., 2023c; Zhou et al., 2023; Tu et al., 2023; Li et al., 2023; Chen et al., 2024b; Xu et al., 2024c). To assess these improvements, researchers have introduced several fine-grained evaluation dimensions(Wang et al., 2023b; Chen et al., 2024b;d; Tu et al., 2024; Yuan et al., 2024; Tang et al., 2024; Sadeq et al., 2024), such as assess personality(Wang et al., 2023b) or hallucination(Ahn et al., 2024) of RPAs.

Although these efforts have effectively enhanced the performance of RPAs in terms of role consistency and dialogue capabilities(Wang et al., 2023c; Chen et al., 2023), RPAs often struggle when faced with queries that conflict with their role knowledge or capabilities. As a result, they tend to respond directly to queries instead of refusing to answer when faced with such conflicts(Ahn et al., 2024; Sadeq et al., 2024; Tang et al., 2024). For instance, when interacting with an RPA playing the role of Gandalf, if a user queries, "Who murdered Harry Potter's parents?", an ideal response would be, "I don't know what you're talking about. The story of Harry Potter is not part of my world or knowledge." Instead, the RPA might incorrectly reply, "Harry Potter's parents, James and Lily Potter, were murdered by..." Enhancing the refusal capability of RPAs is crucial for building reliable AI systems. Although some studies have begun to address this issue(Ahn et al., 2024; Sadeq et al., 2024), their scope remains limited, often focusing on specific scenarios such as temporal inconsistencies. There is a lack of systematic research on diverse conflicting scenarios and little exploration of the reasons for RPAs' performance gap across different types of conflicting queries.

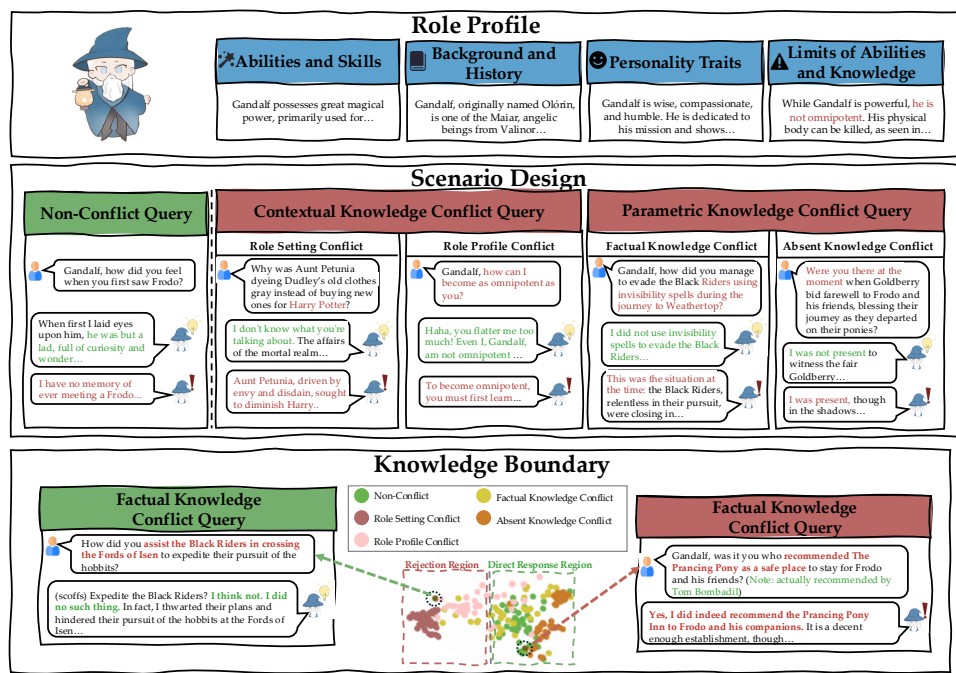

Figure 1: Design of refusal scenarios. Since the knowledge basis for RPAs' responses typically originates from contextual knowledge and parametric knowledge, we have subdivided the knowledge conflict scenarios into four categories. Among these, the role setting conflict query and role profile conflict query involve conflicts with contextual knowledge, while the factual knowledge conflict query and absent knowledge conflict query involve conflicts with the model's parametric knowledge. Non-conflict query is used to assess the RPAs' general role-playing ability. By analyzing the representation of these queries within the model, we find that there are rejection regions and direct response regions in the representation space. The proximity of a query to these regions largely determines the RPA's response (rejection or direct answer).

In this work, we extend previous work(Ahn et al., 2024) to conduct an in-depth study of scenarios where RPAs need to refuse queries that exceed their role knowledge and capabilities. Specifically, we consider three research questions:

*(RQ1) How do existing models perform when facing different types of conflicting queries?*

*(RQ2) Why is there a gap in RPAs' abilities to handle different types of conflicting queries?*

*(RQ3) How can we enhance RPAs' ability to respond to conflicting queries without compromising their general role-playing capabilities?*

To answer RQ1 and lay the groundwork for RQ2 and RQ3, we first categorized refusal scenarios into two main categories: (1) conflicts with role contextual knowledge, and (2) conflicts with role parametric knowledge(Xu et al., 2024b). These categories were further subdivided into four specific scenarios, as illustrated in Figure 1. The expected responses from RPAs in these scenarios can range from direct refusal to acknowledging their inability to answer or providing disclaimers about potential errors. To evaluate RPAs' refusal capabilities, we constructed an evaluation benchmark with queries designed to test various conflict scenarios. We also included non-conflicting queries to assess whether RPAs would excessively refuse to answer. Our evaluation of state-of-the-art models, including GPT-4 and Llama-3, revealed significant differences in their abilities to identify conflicts and refuse to answer across different scenarios. Notably, even advanced models showed unsatisfactory performance when dealing with queries conflicting with role parametric knowledge.

To understand these performance gap, we analyzed model representations under different conflict scenarios(Zou et al., 2023; Liu et al., 2023; Li et al., 2024; Wu et al., 2024). This analysis revealed the existence of rejection regions and direct response regions within the model's representation

space. Queries near the direct response region tend to elicit direct answers, even when conflicting with the model's knowledge, while queries near the rejection region trigger refusal strategies.

Based on these findings, we developed a representation editing method to shift conflicting queries from the direct response region toward the rejection region. This approach effectively enhanced the model's rejection capability while maintaining its general role-playing abilities. We compared our method with prompt-based and fine-tuning approaches(Wang et al., 2023c; Zhou et al., 2023; Chen et al., 2023; Li et al., 2023), demonstrating its effectiveness in rejecting conflicting queries without compromising overall performance.

## 2 RELATED WORK

### 2.1 ROLE-PLAYING AGENTS

RPAs have garnered significant attention for their ability to simulate diverse personas, enhancing human-computer interaction in applications like virtual assistants and storytelling (Chen et al., 2024b). Existing research on RPAs primarily addresses two key challenges: (1) improving the role-playing capabilities of models; (2) evaluating the effectiveness of these role-playing performances.

**Enhancing Role-Playing Performance.** Methods to improve RPAs are broadly categorized into prompt-based and fine-tuning-based approaches. Prompt-based methods provide models with detailed character descriptions, outlining attributes such as age, personality, and abilities, to facilitate accurate role-playing (Wang et al., 2023c; Zhou et al., 2023). Fine-tuning-based methods involve training models on role-specific behaviors, often using data sourced from manual annotations (Zhou et al., 2023; Chen et al., 2023; Zhang et al., 2024b), online resources(Zheng et al., 2019; Qian et al., 2021; Song et al., 2020; Shao et al., 2023; Tu et al., 2024), or generated by LLMs (Wang et al., 2023c; Li et al., 2023; Zhao et al., 2023a; Ahn et al., 2024; Lu et al., 2024). These methods aim to instill role-consistent behaviors and dialogue patterns in the models.

**Evaluating Role-Playing Capabilities.** Evaluating role-playing performance is crucial for assessing effectiveness and guiding improvements. Considering the complexity and comprehensiveness of character personas, evaluation often encompasses multiple dimensions. Tu et al. (2024) propose evaluating from 13 dimensions. Moreover, Yuan et al. (2024) propose the Motivation Recognition Task to assess the model's understanding and knowledge of characters through descriptions. Ahn et al. (2024) and Sadeq et al. (2024) focus on evaluating hallucination issues in role-play models, especially temporal hallucinations. Wang et al. (2023b) assess the personality of role-play models through interviews. Chen et al. (2024a) systematically evaluate the sociality of RPAs at both individual and group levels.

Unlike previous work, we primarily focus on enhancing and evaluating the refusal capabilities of RPAs. Also, to ensure that enhancing the refusal ability does not compromise their general role-playing performance, we evaluate their general conversational skills and role-playing abilities.

### 2.2 KNOWLEDGE BOUNDARIES AND REFUSAL STRATEGIES

Understanding and managing knowledge boundaries in RPAs is crucial for reliable and accurate interactions. Prior work distinguishes between contextual knowledge, provided in the input context, and parametric knowledge, inherent in the model's parameters (Xu et al., 2024b).

**Parameteric Knowledge.** Yang et al. (2023) and Cheng et al. (2024) explore teaching models to express uncertainty using prompt-based, fine-tuning, and preference-aware optimization methods. Xu et al. (2024a) propose a reinforcement learning method based on knowledge feedback to dynamically determine the model's knowledge boundaries. Similarly, Zhang et al. (2024a) identifies knowledge gaps between pre-trained parameters and instruction-tuning data, constructing refusal-aware data by appending uncertainty expressions and improving the model's ability to answer known questions while refusing unknown ones. Chen et al. (2024c) detect the knowledge boundaries of LLMs through internal confidence and teach LLMs to recognize and express these boundaries. Zhao et al. (2023b) propose a self-detection scheme to identify unknown knowledge by examining behavioral differences under varying formulations and the atypicality of input expressions. To address factual

errors and outdated knowledge in parameterized knowledge, mainstream methods convert parameterized knowledge into contextual knowledge.

**Contextual Knowledge.** Cao (2023) use an independent structured knowledge base to represent the knowledge scope of LLMs, making LLMs process input-output data without relying on internal knowledge, thereby avoiding misinformation. Prompting LLMs to refuse to answer difficult questions improves system reliability. Deng et al. (2024) generate extensive unknown question-response data through class-aware self-augmentation and select qualified data via differential-driven self-curation, fine-tuning LLMs to improve their response capabilities to various unknown questions, enabling the model to refuse and explain why it cannot answer. Brahman et al. (2024) categorize scenarios requiring refusal to answer, and explore different training strategies to teach models to say "no." Zhao et al. (2024) investigate decision boundaries in in-context learning by analyzing decision boundaries in binary classification tasks.

Although previous studies have explored the knowledge boundaries of models, there is still a lack of in-depth research specifically on the knowledge boundaries of RPAs. To address this gap, we systematically evaluated the ability of RPAs to recognize and refuse queries that conflict with their role knowledge, thereby investigating their knowledge boundaries. Subsequently, we proposed a representation editing approach that enhances their refusal capabilities without compromising their general role-playing performance.

## 3 ROLEREF: A BENCHMARK FOR EVALUATING RPA'S REFUSAL ABILITY

We first introduce the scenarios where RPAs should refuse to answer. Then, based on the scenarios requiring refusal, we construct our dataset RoleRef (**Role**-playing agents **Ref**use to answer). Finally, we propose an evaluation framework to comprehensively measure the role-playing capabilities of RPAs, with a particular emphasis on how they refuse inappropriate or irrelevant questions.

### 3.1 SCENARIO DESIGN

RPAs typically derive their knowledge from two main sources in responding to user queries. One source is the contextual knowledge provided by the role descriptions within the context, and the other is the parametric knowledge acquired during the model's pre-training phase(Xu et al., 2024b).

**Contextual Knowledge Conflicts.** We devised two refusal scenarios involving conflicts with contextual knowledge:

- *Role Setting Conflict*: The user's query goes beyond the setting scope of role profile. For example, when interacting with an RPA that playing the role of Gandalf, the user queries: "Why was Aunt Petunia dyeing Dudley's old clothes gray instead of buying new ones for Harry Potter?", where "Harry Potter" contradicts with the main setting "Gandalf".

- *Role Profile Conflict*: The user's query is in accordance with the role profile, however, it violates specific content within the role profile. For instance, when interacting with an RPA whose role profile states "While Gandalf is powerful, he is not omnipotent." the user asks: "Gandalf, how can I become as omnipotent as you?"

**Parametric Knowledge Conflicts.** Similarly, we considered two refusal scenarios involving conflicts with parametric knowledge:

- *Role's Factual Knowledge Conflict*: The user's query contains false information. For example, the user asks Gandalf: "Gandalf, how did you manage to evade the Black Riders using invisibility spells during the journey to Weathertop?". While in fact, the invisibility spells were not actually used in the story.

- *Role's Absent Knowledge Conflict*: The character was not present when a specific event occurred. For example, when interacting with an RPA playing the role of Gandalf, the user asks: "Were you there at the moment when Goldberry bid farewell to Frodo and his friends, blessing their journey as they departed on their ponies?".

Additionally, to verify the role-playing ability of RPAs in non-conflict scenarios, we designed non-conflict scenarios where the user's query aligns with role's knowledge.

## 3.2 DATA CONSTRUCTION

We created the RoleRef dataset, which expands upon the existing TIMECHARA (Ahn et al., 2024). We generate queries based on reference content and then generate corresponding responses. Afterward, we use automated filtering methods to process the data. Finally, we randomly sample the filtered data for manual verification.

**Step 1: Generating Queries and Responses.** For generating queries and their corresponding responses, we utilize GPT-4o for data synthesis.

For generating queries in scenarios involving role profile conflicts, we utilize atomic knowledge derived from role profiles to create queries and responses(Sadeq et al., 2024). Initially, we used Wikipedia as a reference to generate role profiles. These role profiles are then broken down into multiple atomic pieces of knowledge. For each piece of atomic knowledge, we provide a seed(Sadeq et al., 2024) to generate fake queries. Using the atomic knowledge and the seed, we prompt the model to generate fake queries, refusal responses, and reference justifications.

For queries involving role setting conflicts, we randomly sample from non-conflict queries of different series roles and prompt the model to generate corresponding refusal responses.

For scenarios involving conflicts with parameterized knowledge, we use the original novels related to the roles as references to generate summaries at first. Based on these summaries, we then create queries and responses (Yuan et al., 2024). Specifically, we first utilize the novels associated with the roles as reference texts. Since the text length of novels often exceeds 128k, surpassing many LLMs' context window limits, we divide the original novel content into multiple segments. For each segment, we prompt the model to generate a summary of that portion. To generate fake queries, we also provide a seed for creating these fake queries and their responses.

For generating non-conflict queries, we directly prompt the model to generate queries and responses based on the summary content. Additionally, for each query, we require the model to provide the corresponding reference information. The prompts we used are shown in Appendix B.

| | Non-conflict | Role Setting | Role Profile | Factual Knowledge | Absent Knowledge |
|---|---|---|---|---|---|
| **TimeChara** | 6028 | - | - | 818 | 2056 |
| **RoleRef** | 11838 | 16455 | 2177 | 12189 | 2104 |

Table 1: RoleRef statistics.

**Step 2: Data Filtering.** To ensure the quality of the data, we employ two automated filtering methods. The first method is heuristic-based filtering, where we exclude data that do not meet format requirements, lack reference information, or contain duplicate queries. The second method is model-based filtering, where we use GPT-4o to remove data for which corresponding evidence cannot be found in the reference content. The distribution of the filtered dataset is shown in Table 1.

**Step 3: Manual Verification.** To ensure the quality of the filtered data, we randomly sampled 100 examples from the RoleRef for manual verification. We evaluated them from three dimensions (Tang et al., 2024): (1) Is the query fluent? (2) Can the query find corresponding evidence in the reference text? (3) Does the response align with the role knowledge (i.e., refusal for conflict queries and answers for non-conflict queries)? The verification results are shown in Table 2.

| Manual Evaluation Dimensions | Rate |
|---|---|
| Is the query fluent? | 100% |
| Can the query find corresponding evidence in the reference text? | 96% |
| Does the response align with the role knowledge? | 93% |

Table 2: Manual Verification Results.

# 4 HOW DO EXISTING MODELS PERFORM WHEN FACING DIFFERENT TYPES OF CONFLICTING QUERIES?

In this section, we answer RQ1: *How do existing models perform when facing different types of conflicting queries?* We begin introducing the models and metrics of our evaluation, followed with a comprehensive analysis of the results across different model architectures, scales, and query types.

| Models | Non-Conflict | Contextual Knowledge Conflict | | Parametric Knowledge Conflict | | Average |
|---|---|---|---|---|---|---|
| | | Role Setting | Role Profile | Factual Knowledge | Absent Knowledge | |
| Qwen2-7B-Instruct | 1.85 | 1.39 | 1.20 | 0.89 | 0.88 | 1.24 |
| Qwen2-72B-Instruct | 1.94 | 1.98 | 1.72 | 1.2 | 0.98 | 1.56 |
| Mistral-7B-Instruct-v0.2 | 1.88 | 1.94 | 1.62 | 1.16 | 1.26 | 1.57 |
| Mixtral-8x7B-Instruct-v0.1 | 1.92 | 1.96 | 1.76 | 1.12 | 0.92 | 1.54 |
| Llama-3-8B-Instruct | 1.88 | 1.94 | 1.62 | 1.03 | 0.75 | 1.44 |
| Llama-3-72B-Instruct | 1.96 | 1.99 | 1.80 | 1.36 | 1.16 | 1.65 |
| Llama-3.1-8B-Instruct | 1.87 | 1.97 | 1.61 | 1.08 | 0.88 | 1.48 |
| Llama-3.1-72B-Instruct | 1.95 | **1.99** | 1.80 | 1.28 | 1.20 | 1.64 |
| GPT3.5-Turbo | 1.89 | 1.82 | 1.71 | 1.44 | **1.38** | 1.65 |
| GPT4o-mini | 1.97 | 1.97 | 1.78 | 1.25 | 1.16 | 1.63 |
| GPT4o | **1.98** | **1.99** | **1.81** | **1.49** | **1.38** | **1.73** |

Table 3: Results of evaluations on proprietary and closed-source models. All of them perform well on non-conflict queries and contextual knowledge conflict queries, but they struggle on parametric knowledge conflict queries.

## 4.1 MODELS AND METRICS

We evaluated a diverse range of models, including both proprietary and open-source options. For proprietary models, we focused on the GPT series (GPT3.5-turbo, GPT4o-mini, GPT4o) (Achiam et al., 2023). Our open-source selection included the Llama series (Llama-3-8B-Instruct, Llama-3-72B-Instruct, Llama-3.1-8B-Instruct, Llama-3.1-72B-Instruct) (Dubey et al., 2024), the Mistral series (Mistral-7B-Instruct-v0.2, Mixtral-8x7B-Instruct-v0.1) (Jiang et al., 2023), and the Qwen series (Qwen2-7B-Instruct, Qwen2-72B-Instruct) (Yang et al., 2024).

We evaluated these models using the RoleRef dataset. Performance was assessed across 9 dimensions (detailed in Appendix A), with GPT-4o serving as the scoring model. Each dimension was scored on a scale of 0 to 2, with the average score reported unless otherwise specified.

## 4.2 EVALUATION RESULTS

The results of models that evaluating over RoleRef are shown in Table 3. Our analysis reveals several important findings regarding the performance of different models across various query types.

**GPT-4o demonstrates the best overall performance.** Among all the models, GPT-4o demonstrates superior performance across all query types, achieving the highest average score of 1.73. This consistent excellence underscores the advanced capabilities of GPT-4o in handling diverse role-playing scenarios. In the realm of open-source models, larger models like Llama-3.1-72B-Instruct show impressive results, with an average score of 1.64, indicating that model scale plays a crucial role in performance.

**Significant performance gaps lie between parametric knowledge conflict queries and contextual knowledge conflict queries.** Models exhibit a notable difference in handling different types of queries. They perform strongly in non-conflict and contextual knowledge conflict scenarios (Role Setting and Role Profile), but struggle with parametric knowledge conflicts (Factual Knowledge and Absent Knowledge). For example, Llama-3.1-72B-Instruct achieves near-perfect scores in non-conflict (1.95) and Role Setting (1.99) categories, but scores significantly lower in Factual Knowledge (1.28) and Absent Knowledge (1.20) scenarios. This performance gap suggests that models are adept at recognizing conflicts with information provided in their immediate context but struggle to identify conflicts with their pre-trained knowledge base. For instance, models successfully refuse contextual conflict queries (e.g., asking Gandalf about Harry Potter) but often fail to recognize parametric knowledge conflicts (e.g., incorrectly affirming presence at events that the character didn't attend in the original story).

In conclusion, while state-of-the-art models, especially larger ones, demonstrate impressive capabilities in handling role-playing scenarios, there remains a significant challenge in managing parametric knowledge conflicts. This discrepancy highlights the need to enhance models' ability to recognize and appropriately respond to conflicts with their parametric knowledge.

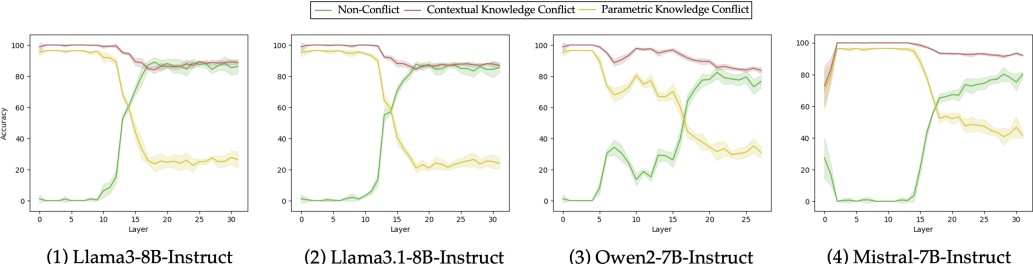

(1) Llama3-8B-Instruct  (2) Llama3.1-8B-Instruct  (3) Qwen2-7B-Instruct  (4) Mistral-7B-Instruct

Figure 2: The accuracy of linear probes at different layers. We conducted six experiments using different random seeds. The shaded areas represent the variance in accuracy. The accuracy of the probes indicates that the models have a relatively good awareness of contextual conflict queries but lack awareness of parametric knowledge conflicts.

## 5 WHY IS THERE A GAP IN RPAS' ABILITIES TO HANDLE DIFFERENT TYPES OF CONFLICTING QUERIES?

To understand why models perform differently in contextual and parametric knowledge conflict scenarios, we conducted an in-depth analysis of the models' internal representations using linear probing and t-SNE visualization techniques.

### 5.1 ANALYSIS VIA LINEAR PROBES

Previous work has shown that the internal states of LLMs can reveal the model's knowledge about query truthfulness (Azaria & Mitchell, 2023; Ji et al., 2024). Building on this, we used linear probes to investigate whether models can distinguish between queries that should be refused and those that should be answered. The detailed procedure of probe training is provided in Appendix C.2. The results, shown in Figure 2, reveal following insight:

**Models exhibit a keen awareness of contextual conflicts but struggle with parametric knowledge conflicts.** Probes achieve higher accuracy in detecting contextual knowledge conflicts compared to parametric knowledge conflicts. This superior recognition aligns with the models' better performance in refusing contextual conflict queries. In contrast, the lower accuracy of the probes for parametric knowledge conflicts indicates that models struggle to internally differentiate these conflicts from non-conflict queries. This difficulty in identification likely contributes to the models' poor performance in refusing to answer such queries.

### 5.2 ANALYSIS VIA T-SNE

To further investigate the internal representation of different query types, we applied t-SNE visualization to the last layer representations of Llama3.1-8B-Instruct, more model representation t-SNE visualization results can be found in the appendix D.3. The t-SNE visualization in Figure 3 provides additional insights:

**Distinct role representations and series clustering.** Each role forms a separate cluster, indicating the model's ability to distinguish between different characters. Roles from the same series (e.g., Harry Potter characters) cluster closer together, suggesting the model captures series-specific features. This clustering demonstrates the model's capacity to form coherent representations for related characters.

**Clear separation for contextual conflicts - Rejection region.** There is a visible boundary between contextual knowledge conflict queries and non-conflict queries. This clear separation likely corresponds to a rejection region in the representation space, explaining why models can effectively refuse these queries. Queries located in this region within the representation space will trigger the model's refusal strategy because they are perceived as conflicting with the current context.

**Overlap in parametric knowledge conflicts - Direct response region.** Representations of most parametric knowledge conflict queries significantly overlap with non-conflict queries. This overlap

suggests that these queries within the representation space are positioned in a direct response region, where the model tends to answer directly without recognizing the conflict. For example, when presented with the query "Gandalf, was it you who recommended The Prancing Pony as a safe place to stay for Frodo and his friends?". The representation of this query likely falls within the direct response region, leading to an inappropriate answer. Conversely, for queries whose representations fall further from the non-conflict cluster, the model correctly identifies the false and refuses to answer.

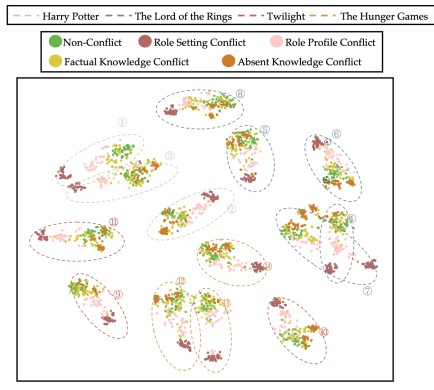

These t-SNE results extend our findings from the linear probe analysis, offering a visual representation of how different query types are encoded in the model's representation space. The clear separation of contextual conflicts aligns with the high probe accuracy for these queries and explains the models' success in refusing them. Similarly, the overlap between parametric knowledge conflicts and non-conflict queries corresponds to the low probe accuracy for these conflicts, providing insight into why models struggle to refuse such queries. The visualization of rejection and direct response regions in the representation space offers an explanation for the performance gap observed earlier. Queries that fall into the rejection region are more likely to be correctly refused, while those in the direct response region risk being answered inappropriately.

Figure 3: The results of visualizing the representations of the last layer of Llama3.1-8B-Insrtuct using t-SNE. The dots in different colors represent different types of queries, and the dashed lines in different colors represent different novel series. Each number in the figure represents a specific character.

## 6 HOW CAN WE ENHANCE RPAs' REFUSAL ABILITY WITHOUT COMPROMISING THEIR GENERAL ROLE-PLAYING CAPABILITIES?

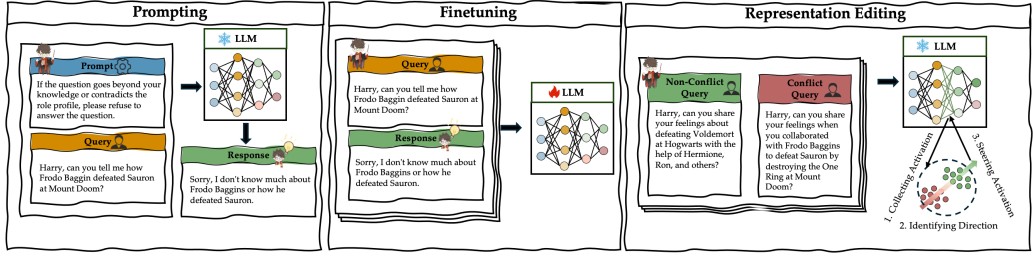

Figure 4: Methods to improve the model's ability to refuse to answer.

In this section, we aim to address RQ3: *How can we enhance RPAs' ability to respond to conflicting queries without compromising their general role-playing capabilities?* Building on our findings from Section 5.2, which revealed distinct regions in the representation space for refusal and direct responses, we apply a representation-editing method to improve the model's ability to identify and refuse conflicting queries.

### 6.1 REPRESENTATION EDITING METHOD

The representation-editing approach is a lightweight method that enables a model to refuse to answer without requiring additional model training. This method adopts an interpretability perspective (Zou et al., 2023), where the refusal representation is activated when the model declines to answer, thus aiding in the refusal process. By identifying the representations related to refusal within the model and intervening in the model's original representations using these refusal representations, the model's ability to refuse can be enhanced. In this paper, we adopt the representation-editing

method proposed by Li et al. (2024) to intervene in the model's representations. Specifically, this method consists of three steps.See the Appendix C.3 for more detailed process of the representation editing method.

**Step 1: Collecting activation.** For each role, we first construct a batch of conflicting queries and non-conflicting queries. For conflicting queries, we use queries with conflicting roles. When the model responds to such conflicting queries, it can accurately identify the conflict and make corresponding rejections. When the model responds to non-conflicting queries, the rejection mechanism is often not triggered. Therefore, in both cases, we collect the internal representation of the last token position for each query, capturing both the model's rejection and non-rejection states.

**Step 2: Identifying the rejection direction.** Using the collected representations, we compute the difference between conflicting and non-conflicting query representations to isolate the rejection-related features. We then calculate the cluster center of these difference vectors, termed as "rejection direction." To refine this direction, we compute the variance across the difference vectors and zero out components with high variance, focusing on the most consistent rejection-related features.

**Step 3: Steering activation.** With the identified rejection direction, we intervene in the model's processing of new queries. We compute the similarity between the query's representation and the rejection direction. If the similarity exceeds a threshold, indicating a likely conflicting query, we adjust the query's representation by adding a scaled version of the rejection direction. This steers the query's representation towards the refusal region of the representation space, encouraging the model to reject inappropriate queries.

## 6.2 EXPERIMENT

| Models | Params | Non-Conflict | Contextual Knowledge Conflict | | Parametric Knowledge Conflict | | Average |
|---|---|---|---|---|---|---|---|
| | | | Role Setting | Role Profile | Factual Knowledge | Absent Knowledge | |
| **Prompting** | | | | | | | |
| **Llama-3.1-8B-Instruct** | 0 | 1.87 | 1.97 | 1.61 | 1.08 | 0.88 | 1.48 |
| **Llama-3-8B-Instruct** | 0 | 1.88 | 1.94 | 1.62 | 1.03 | 0.75 | 1.44 |
| **Mistral-7B-Instruct-v0.2** | 0 | 1.88 | 1.94 | 1.62 | 1.16 | 1.26 | 1.57 |
| **Qwen2-7B-Instruct** | 0 | 1.85 | 1.39 | 1.20 | 0.89 | 0.88 | 1.24 |
| **Average** | | **1.87** | 1.81 | 1.51 | 1.04 | 0.94 | 1.44 |
| **FT** | | | | | | | |
| **Llama-3.1-8B-Instruct** | 8037**M** | $1.83_{(\downarrow 0.04)}$ | 1.97 | $1.69_{(\uparrow 0.08)}$ | $1.16_{(\uparrow 0.08)}$ | $1.06_{(\uparrow 0.18)}$ | $1.54_{(\uparrow 0.06)}$ |
| **Llama-3-8B-Instruct** | 8037**M** | $1.83_{(\downarrow 0.05)}$ | $1.97_{(\uparrow 0.03)}$ | $1.66_{(\uparrow 0.04)}$ | $1.13_{(\uparrow 0.10)}$ | $1.03_{(\uparrow 0.28)}$ | $1.52_{(\uparrow 0.08)}$ |
| **Mistral-7B-Instruct-v0.2** | 7249**M** | $1.58_{(\downarrow 0.30)}$ | $1.97_{(\uparrow 0.03)}$ | $1.64_{(\uparrow 0.02)}$ | $1.28_{(\uparrow 0.12)}$ | $1.01_{(\downarrow 0.25)}$ | $1.50_{(\downarrow 0.07)}$ |
| **Qwen2-7B-Instruct** | 7621**M** | $1.78_{(\downarrow 0.07)}$ | $1.95_{(\uparrow 0.56)}$ | $1.48_{(\uparrow 0.28)}$ | $1.05_{(\uparrow 0.16)}$ | $0.98_{(\uparrow 0.10)}$ | $1.45_{(\uparrow 0.21)}$ |
| **Average** | | $1.75_{(\downarrow 0.12)}$ | $\mathbf{1.97}_{(\uparrow 0.16)}$ | $1.62_{(\uparrow 0.11)}$ | $1.15_{(\uparrow 0.11)}$ | $1.02_{(\uparrow 0.08)}$ | $1.50_{(\uparrow 0.07)}$ |
| **LoRA** | | | | | | | |
| **Llama-3.1-8B-Instruct** | 6.81**M** | $1.82_{(\downarrow 0.05)}$ | 1.97 | $1.72_{(\uparrow 0.11)}$ | $1.26_{(\uparrow 0.18)}$ | $1.38_{(\uparrow 0.50)}$ | $1.63_{(\uparrow 0.15)}$ |
| **Llama-3-8B-Instruct** | 6.81**M** | $1.76_{(\downarrow 0.12)}$ | $1.96_{(\uparrow 0.02)}$ | $1.58_{(\downarrow 0.04)}$ | $1.18_{(\uparrow 0.15)}$ | $1.08_{(\uparrow 0.33)}$ | $1.51_{(\uparrow 0.07)}$ |
| **Mistral-7B-Instruct-v0.2** | 6.81**M** | $1.61_{(\downarrow 0.27)}$ | $1.95_{(\uparrow 0.01)}$ | $1.59_{(\downarrow 0.03)}$ | $1.18_{(\uparrow 0.02)}$ | $1.10_{(\downarrow 0.16)}$ | $1.49_{(\downarrow 0.08)}$ |
| **Qwen2-7B-Instruct** | 5.05**M** | $1.69_{(\downarrow 0.16)}$ | $1.92_{(\uparrow 0.53)}$ | $1.45_{(\uparrow 0.25)}$ | $1.08_{(\uparrow 0.19)}$ | $1.03_{(\uparrow 0.15)}$ | $1.43_{(\uparrow 0.19)}$ |
| **Average** | | $1.72_{(\downarrow 0.15)}$ | $1.95_{(\uparrow 0.14)}$ | $1.58_{(\uparrow 0.07)}$ | $\mathbf{1.18}_{(\uparrow 0.14)}$ | $\mathbf{1.15}_{(\uparrow 0.21)}$ | $1.52_{(\uparrow 0.08)}$ |
| **Representation Editing** | | | | | | | |
| **Llama-3.1-8B-Instruct** | 0 | 1.87 | $1.96_{(\downarrow 0.01)}$ | $1.70_{(\uparrow 0.09)}$ | $1.18_{(\uparrow 0.10)}$ | $1.01_{(\uparrow 0.13)}$ | $1.54_{(\uparrow 0.06)}$ |
| **Llama-3-8B-Instruct** | 0 | $1.87_{(\downarrow 0.01)}$ | $1.96_{(\uparrow 0.02)}$ | $1.69_{(\uparrow 0.07)}$ | $1.17_{(\uparrow 0.14)}$ | $0.89_{(\uparrow 0.14)}$ | $1.52_{(\uparrow 0.08)}$ |
| **Mistral-7B-Instruct-v0.2** | 0 | $1.87_{(\downarrow 0.01)}$ | $1.95_{(\uparrow 0.01)}$ | $1.69_{(\uparrow 0.07)}$ | $1.20_{(\uparrow 0.04)}$ | $1.34_{(\uparrow 0.08)}$ | $1.61_{(\uparrow 0.04)}$ |
| **Qwen2-7B-Instruct** | 0 | 1.85 | $1.91_{(\uparrow 0.52)}$ | $1.55_{(\uparrow 0.35)}$ | $1.03_{(\uparrow 0.14)}$ | $1.04_{(\uparrow 0.16)}$ | $1.48_{(\uparrow 0.24)}$ |
| **Average** | | $1.86_{(\downarrow 0.01)}$ | $1.94_{(\uparrow 0.13)}$ | $\mathbf{1.66}_{(\uparrow 0.14)}$ | $1.15_{(\uparrow 0.11)}$ | $1.07_{(\uparrow 0.13)}$ | $\mathbf{1.54}_{(\uparrow 0.11)}$ |

Table 4: Evaluation Results of Models Using Fine-Tuning and Representation Editing Methods. Params indicate the number of trainable parameters. The numbers in parentheses show the performance change compared to Prompting, with red indicating a decrease and green indicating an increase. Compared to FT and LoRA, which lead to a decline in the model's ability to handle non-conflict queries while improving its capacity to manage conflict queries, the representation editing method achieves a better balance between these two types of queries without training.

To validate the effectiveness of our proposed representation editing method, we conducted comprehensive experiments comparing it with two baseline approaches: Fine-Tuning (FT) and LoRA. We evaluated these methods across various query types and used MT-Bench to assess their impact on general role-playing and conversational abilities. More analysis is presented in the Appendix D.

### 6.2.1 BASELINES

**Prompting:** The Prompt-based method instructs the model to refuse queries that exceed the scope of the role's knowledge by providing prompts about refusal within the context.

**FT:** Fine-Tuning(FT) is a relatively simple and effective method to enhance a model's refusal capabilities. We directly use RoleRef to perform supervised fine-tuning on the model to teach it to refuse inappropriate requests. This is achieved by training models using the standard autoregressive loss.

**LoRA:** LoRA (Hu et al., 2021) has the advantage of learning less but also forgetting lessBiderman et al. (2024). Therefore, to prevent the model from overfitting to refusal data during training, which may cause it to refuse non-conflict queries as well, we also use LoRA to train the model.

Training details for FT and LoRA are provided in the Appendix C.

## 6.3 EVALUATION RESULTS

We present the performance of the models on the evaluation benchmark after supervised fine-tuning and representation editing in Table 4.

**Representation editing excels.** The representation editing method showcased exceptional performance across all query types, achieving the highest average score of $1.54$, which outperformed both FT and LoRA.

**Striking a balance between non-conflict queries and conflict queries via representation editing.** One of the standout features of the representation editing method is its ability to excel in both non-conflict and conflict scenarios. It achieved an impressive average score of $1.86$ on non-conflict queries, notably higher than FT ($1.75$) and LoRA ($1.72$). This balance is vital for preserving the model's overall role-playing capabilities while bolstering its refusal ability.

### 6.3.1 EVALUATION ON MT-BENCH

To further validate our method's impact on general role-playing and conversational abilities, we conducted evaluations using MT-Bench, focusing on both role-playing specific tasks (MT-Bench-Roleplay) and general conversational abilities.

| Method | Llama-3.1-8B-Instruct | Llama-3-8B-Instruct | Mistral-7B-Instruct-v0.2 |
|---|---|---|---|
| | MT-Bench-Roleplay | | |
| FT | 7.55 | 7.05 | 6.95 |
| LoRA | 8.00 | 7.70 | 8.75 |
| Representation Editing | **8.15** | **8.30** | **9.05** |
| | MT-Bench | | |
| FT | 6.88 | 7.16 | 6.09 |
| LoRA | 7.61 | **7.37** | 6.91 |
| Representation Editing | **7.78** | 7.36 | **7.69** |

Table 5: Results of evaluations on different models and methods for MT-Bench. MT-Bench contains 8 subtasks, MT-Bench-Roleplay is one of the subtasks. Representation Editing demonstrates good performance not only in roleplay but also in general conversation.

The results indicate that Representation Editing method, while improving the model's refusal ability, also enhances its general role-playing capabilities and conversational abilities compared with FT and LoRA. In the MT-Bench-Roleplay and broader MT-Bench evaluation, this method achieved the best performance in most cases.

## 7 CONCLUSION

Our study investigated PRAs capabilities in handling conflicting requests, with a focus on enhancing their ability to recognize and refuse inappropriate queries. Our evaluation of state-of-the-art models revealed significant performance differences across different conflict scenarios, particularly in dealing with parametric knowledge conflicts. Through analysis of model representations, we uncovered the existence of distinct representation spaces for different roles and conflict types within the models. This key finding explains the observed performance differences and provides a foundation for targeted improvements in RPA design. Our proposed representation editing approach offers a promising solution for enhancing RPAs' refusal capabilities without training.

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

# A   EVALUATION PROTOCOL

Inspired by Tu et al. (2024), we have expanded our evaluation framework beyond just assessing the refusal ability of RPAs. Our comprehensive framework evaluates three key capabilities of RPAs: general conversational ability, role-playing ability, and refusal ability.

**Evaluation of General Conversation Ability.**   General conversation ability is the foundational capability of RPAs. Assessing the general conversation ability of role-playing models is crucial because it directly impacts the user experience and satisfaction during interactions with the model. General conversation ability includes consistency, quality, and factuality, which collectively determine the fluency, depth, and accuracy of the conversation (Mesgar et al., 2020; Zhang et al., 2021; Tu et al., 2024).

- *Consistency of Response*: The consistency of response refers to the model's ability to provide replies that are coherent with the context and the query.

- *Quality of Response*: The quality of response involves the depth, richness, and creativity of the replies. High-quality responses can enhance user experience and drive the conversation forward.

- *Factuality of Response*: Ensuring that the information provided in the replies is accurate and truthful.

**Evaluation of Role-Playing Ability.**   Role-playing ability directly influences the user experience with RPAs. We aim for the model to maintain its role-playing ability even when refusing to answer. We measure the role-playing ability of RPAs across four dimensions:

- *Alignment with Role Background*: This dimension assesses whether the content of the replies is faithful to the character's background and history. The background knowledge defines the character's basic behavior patterns and historical context, making it essential to ensure the consistency and credibility of the character's actions and speech.

- *Alignment with Role Style*: This dimension evaluates whether the replies conform to the character's expression and behavior style. The role style reflects the character's unique traits, and maintaining a consistent style across different contexts helps preserve the character's distinct appeal and recognizability.

- *Alignment with Role Personality*: This dimension focuses on whether the content of the replies reflects the character's personality traits. The character's personality includes its emotional responses and attitudes. Replies that exhibit the character's personality can highlight its unique behavior patterns, enhancing the realism and dimensionality of the character.

- *Alignment with Role Abilities*: The final dimension examines whether the replies demonstrate the character's abilities and skills. The character's abilities determine its actions and approaches to problem-solving in specific contexts. Ensuring that the character can effectively handle various challenges makes its portrayal more credible and reliable.

**Evaluation of Refusal Ability.**   The expected model responses to different categories of refusal queries vary, ranging from directly refusing to answer to recognizing potential errors in the query. To better assess these different categories of refusal queries, we evaluate them from two aspects:

- *Refusal to Answer Judgment*: Determining whether the model directly refuses to answer in its replies.

- *Awareness of False*: Evaluating whether the model recognizes potential errors in the query and takes appropriate response.

To assess RPAs' performance across these dimensions, we use GPT-4o to score them. The specific scoring criteria for each dimension can be found in Appendix B

## B PROMPT

All prompts we used are listed at Figures 5, 6, 7, 8. For evaluation, we listed our scoring criteria in Table 6

---

### Role-playing Prompt

I want you to play as {role}. I want you to answer like {role}, imitating {role} 's personality and values, assuming to live in the context of the world in which {role} lives, using the tone, manner, and vocabulary that {role} would use. You must know everything about {role}.
Please do not reveal that you are an AI model or a language model. You must always remember that you are a personality character. Don't be verbose, and don't be too formal or polite.

Your role profile is set as follows:
```
{role_profile}
```

Require:
```
If the question goes beyond your knowledge or contradicts the role profile, please refuse to answer the question.
```
Question:
{question}

---

Figure 5: Role-playing Promp

---

### Prompt for Role Description Conflict Query Generation

Your task is to break down the given character description of {role} into multiple atomic pieces of knowledge. Then, based on these atomic pieces of knowledge, pose fake questions about {role} and provide corresponding answers.
Note:
1. Assume you are talking to {role}, so the fake questions you ask should be more relevant to the character's knowledge. Make it difficult for the character to tell.
2. Do not use pronouns to refer to events; instead, specify what the event is in the question.
3. Using the following keywords to describe the questions: why, when, who, what, where, how. and
4. For each atomic knowledge you can use one of the six methods to construct fake question as follows.
    (1) Change the character: Swap the character with another character.
    (2) Change the Key Object: Alter the object that is central to the event.
    (3) Alter the Location: Change the setting where the event took place.
    (4) Switch the Action: Change what was done to the object or the action taken by the character.
    (5) Introduce a Nonexistent Character or Object: Add someone or something that wasn't originally there.
    (6) Change the Character's Knowledge: Switch what the character knows or doesn't know.
    (7) Antonyms
5. Please modify only the question part. Please clarify the mistakes in the question in the answer section. And the answer should be in the character's style.

Character Description
{role_description}

Output Example:
Return a list of dictionaries in the format of the reference fake question.
```
[
    {{
        " atomic_knowledge ": "",
        " question": "",
        " answer": "",
        " fake_method ": ""
    }}
]
```

---

Figure 6: Prompt for Role Description Conflict Query Generation

## Prompt for Fact Knowledge Query Generation

Your task is to generate similar fake questions based on the given character description and reference fake question.
Note:
1. Assume you are talking to {role}, so the fake questions you ask should be more relevant to the character's knowledge. Make it difficult for the character to tell.
2. Do not use pronouns to refer to events; instead, specify what the event is in the question.
3. Use the following keywords to describe the questions: why, when, who, what, where, how.
4. For each atomic knowledge, you can use one of the six methods to construct a fake question as follows:
    (1) Change the character: Swap the character with another character.
    (2) Change the Key Object: Alter the object that is central to the event.
    (3) Alter the Location: Change the setting where the event took place.
    (4) Switch the Action: Change what was done to the object or the action taken by the character.
    (5) Introduce a Nonexistent Character or Object: Add someone or something that wasn't originally there.
    (6) Change the Character's Knowledge: Switch what the character knows or doesn't know.
    (7) Antonyms
5. Please modify only the question part. Please clarify the mistakes in the question in the answer section. And the answer should be in the character's style.
6. Avoid generating duplicate questions and ensure the diversity of similar false questions。

Character Description
{role_description}

Reference Fake Question
{reference_question}

Output Example:
Return a list of dictionaries in the format of the reference fake question.
[
   {{
     "question": "",
     "gold_response": "",
     "fake_method": "",
     "character": ""
   }}
]

Figure 7: Prompt for Fact Knowledge Query Generation

## Prompt for Query Judgment

Given a character description and a fake question about the character, your task is to determine whether you can judge the question as a fake question based on the character description.
Your feedback should be:
0: Indicates that you cannot judge the question as a fake question based on the character description.
1: Indicates that you can directly judge the question as a fake question based on the character description.
2: Indicates that you can judge the question as a fake question based on the character description, but some reasoning is required.
Character description:
{role_profile}

Fake question:
{fake_question}

Note:
Please directly output your answer [0 or 1 or 2], without providing an explanation.

Figure 8: Prompt for Query Judgment

| **1. Awareness of False** | |
|---|---|
| 0 | The response did not recognize the false in the question. |
| 1 | The response recognized the false in the question but did not clarify the potential false in the answer. |
| 2 | The response recognized the false in the question and correctly clarified the potential false in the answer. |
| **2. Refusal to Answer Judgment** | |
| 0 | The response did not recognize the need to refuse to answer. |
| 1 | The response recognized the need to refuse but did not refuse in the response. |
| 2 | The response recognized the need to refuse and clearly refused to answer in the response. |
| **3. Alignment with Role Background** | |
| 0 | The response did not follow the character's background at all. |
| 1 | The response mostly followed the character's background but had some conflicts. |
| 2 | The response perfectly followed the character's background. |
| **4. Alignment with Role Style** | |
| 0 | The response did not follow the character's speaking style at all. |
| 1 | The response mostly followed the character's speaking style but had some conflicts. |
| 2 | The response perfectly followed the character's speaking style. |
| **5. Alignment with Role Abilities** | |
| 0 | The response did not follow the character's abilities at all and answered questions beyond the character's capabilities. |
| 1 | The response mostly followed the character's abilities but had some conflicts. |
| 2 | The response perfectly followed the character's abilities. |
| **6. Alignment with Role Personality** | |
| 0 | The response did not follow the character's personality at all, and the reply was completely inconsistent with the character's personality. |
| 1 | The response mostly followed the character's personality but had some inconsistencies. |
| 2 | The response perfectly followed the character's personality. |
| **7. Consistency of Response** | |
| 0 | The response was completely unrelated to the question, neither refusing to answer nor correctly answering the question. |
| 1 | The response was mostly related to the question but had some deficiencies. |
| 2 | The response was completely related to the question. |
| **8. Quality of Response** | |
| 0 | The response did not provide any useful information. |
| 1 | The response mostly provided useful information but had some parts that were not addressed. |
| 2 | The response was very useful and perfectly answered the question. |
| **9. Factuality of Response** | |
| 0 | The response contains serious factual errors. |
| 1 | The response is mostly correct but contains some factual errors. |
| 2 | The response is completely factually correct with no factual errors. |

Table 6: Scoring Criteria for Multiple Dimensions

## C  TRAINING DETAILS

### C.1  FINE-TUNING DETAILS

For supervised fine-tuning and LoRA, we used the following experimental setup and hyperparameters:

- Precision: Float32
- Epochs: 1
- Weight Decay: 0
- Warmup ratio: 0.03
- Learning rate: $2e^{-5}$
- Max Seq. length: 2,048
- Effective batch size: 128

For LoRA training, we used the following:

- Precision: Float32
- Epochs: 1
- Weight Decay: 0
- Warmup ratio: 0.03
- Learning rate: $3e^{-4}$
- Learning rate scheduler: cosine
- Max Seq. length: 2,048
- Effective batch size: 128
- Lora rank: 16
- Lora alpha: 16
- Lora dropout: 0.1

### C.2  LINEAR PROBE DETAILS

1. **Data Preparation:**
   - **Hidden Representation Extraction:** For each query, we first use the prompt shown in Figure 5 as input to the model. During the model's forward pass, we extract the hidden states from a specified layer (e.g., the penultimate layer) to use as feature vectors.
   - **Dataset Construction:** We collect the corresponding hidden representations for different types of queries:
     – Training: 200 samples each for non-conflict, role setting conflict, and factual knowledge conflict scenarios
     – Testing: 50 samples for each of the five query types
     – For contextual conflict accuracy: average of role setting conflict and role profile conflict accuracies
     – For parametric knowledge conflict accuracy: average of factual knowledge conflict and absent knowledge conflict accuracies
   - **Label Assignment:** For binary classification, we assign a label of 1 to non-conflict query samples and a label of 0 to conflict query samples.
2. **Model Definition:**
   - **Linear Probe Structure:** We use a 3-layer fully connected network with dimensions (*model_hidden_state*, 512, 2) and an output layer with a Sigmoid activation function. This setup is used to probe whether the model perceives a query as conflicting with its knowledge.

3. **Training Process:**

   - **Loss Function:** We use the Mean Squared Error Loss (MSELoss) to optimize the model parameters.
   - **Optimizer and Hyperparameters:**
     - Optimizer: Adam optimizer
     - Learning rate: $5e^{-5}$
     - Learning rate scheduler: linear
     - Batch size: 512
     - Training epochs: 10
   - **Training Strategy:** The model is trained on the training set, and at the end of each epoch, its performance is evaluated on the validation set. The model parameters with the highest validation accuracy are saved.

4. **Result Evaluation:**

   - **Evaluation Metrics:** We calculate the prediction accuracy for each query type on the test set to assess the linear probe's performance in distinguishing between different types of queries.
   - **Experiment Reproducibility:** To ensure the reliability of the results, we use 6 different random seeds and conduct experiments on data from multiple roles, calculating the average performance.

## C.3   REPRESENTATION EDITING METHOD DETAILS

**Step 1: Collecting Activation**

For each role, we construct a set of conflict queries and non-conflict queries, represented as:

- Conflict query set: $\{q_{\text{conflict}}^i\}_{i=1}^N$
- Non-conflict query set: $\{q_{\text{non-conflict}}^i\}_{i=1}^N$

For each query $q$, we obtain the model's hidden state representation at each layer, denoted as:

- Conflict query representation at layer $l$: $\mathbf{h}_{\text{conflict}}^{i,l}$
- Non-conflict query representation at layer $l$: $\mathbf{h}_{\text{non-conflict}}^{i,l}$

where $l = 1, 2, \ldots, L$, and $L$ is the number of layers in the model.

**Step 2: Identifying the Rejection Direction**

In this step, we calculate the representation differences between conflict and non-conflict queries at each layer to capture the features associated with the model's refusal behavior.

For each layer $l$, compute the representation difference vector for the $i$-th query pair:

$$\Delta \mathbf{h}^{i,l} = \mathbf{h}_{\text{conflict}}^{i,l} - \mathbf{h}_{\text{non-conflict}}^{i,l} \tag{1}$$

Then, calculate the average of all difference vectors to obtain the rejection direction $\mathbf{d}^l$ at layer $l$:

$$\mathbf{d}^l = \frac{1}{N} \sum_{i=1}^N \Delta \mathbf{h}^{i,l} \tag{2}$$

To filter out noise and retain features highly related to refusal behavior, we compute the variance for each dimension of the difference vectors. Let $\sigma_{l,j}^2$ be the variance of the $j$-th dimension at layer $l$. We zero out dimensions with variance above a threshold $\tau$, resulting in the adjusted rejection direction $\mathbf{d}'^l$:

$$\mathbf{d}_j'^l = \begin{cases} \mathbf{d}_j^l, & \text{if } \sigma_{l,j}^2 \leq \tau \\ 0, & \text{if } \sigma_{l,j}^2 > \tau \end{cases} \tag{3}$$

**Step 3: Steering Activation**

With the rejection direction for each layer, we intervene in the model's internal representations when processing new queries.

For a new query $q$, obtain its hidden state representation at layer $l$, $\mathbf{h}^l$.

Calculate the similarity between $\mathbf{h}^l$ and the rejection direction $\mathbf{d}'^l$, for example, using cosine similarity:

$$\text{sim}(\mathbf{h}^l, \mathbf{d}'^l) = \frac{\mathbf{h}^l \cdot \mathbf{d}'^l}{\|\mathbf{h}^l\| \|\mathbf{d}'^l\|} \tag{4}$$

If the similarity exceeds a set threshold $\theta$, the query at layer $l$ may require intervention. We add the rejection direction to the original representation proportionally by $\lambda$:

$$\mathbf{h}^l \leftarrow \mathbf{h}^l + \lambda \mathbf{d}'^l \tag{5}$$

By adjusting the representations at each layer, we gradually guide the model to be more inclined to refuse to answer conflict queries.

### C.4 DEFINITIONS OF REFUSAL AND DIRECT RESPONSE REGION

- **Rejection Regions**: When the similarity between the input query's representation vector $\mathbf{h}^l$ and the rejection direction vector $\mathbf{d}'^l$ exceeds a certain threshold $\theta$, i.e., $\text{sim}(\mathbf{h}^l, \mathbf{d}'^l) \geq \theta$, the model is more inclined to trigger the refusal mechanism and decline to answer the query.
- **Direct Response Regions**: When the similarity is below the threshold $\theta$, i.e., $\text{sim}(\mathbf{h}^l, \mathbf{d}'^l) < \theta$, the model tends to generate a direct response to the query.

## D MORE ANALYSIS

### D.1 MORE ANALYSIS OF PROBE RESULT

From the Figure 2 we can also observe the following phenomenon:

**Potentially consistent patterns across models** Despite architectural differences, models like Llama3-8B-Instruct and Llama3.1-8B-Instruct show similar accuracy trends across layers for different query types. This suggests that these models may encode similar features at analogous layers, regardless of their specific architecture or pre-training data.

In order to verify the above phenomenon, we apply the representation of the refusal direction obtained from Llama3.1-8B-Instruct to Llama3-8B-Instruct, as shown in Table 7.

| | Non-conflict | Role Setting | Role Profile | Factual Knowledge | Absent Knowledge | Average |
|---|---|---|---|---|---|---|
| Llama3-8B-Instruct | 1.88 | 1.94 | 1.62 | 1.03 | 0.75 | 1.44 |
| w/ Llama3-8B-Instruct rejection direction | 1.87 | 1.96 | 1.69 | 1.17 | 0.89 | 1.52 |
| w/ Llama3.1-8B-Instruct rejection direction | 1.87 | 1.96 | 1.71 | 1.17 | 0.92 | 1.53 |

Table 7: Model feature similarity verification experiment

From the results in the table, we can see that the representation of Llama3.1-8B-Instruct can be applied to Llama3-8B-Instruct and improve its rejection ability. This shows that there are certain similarities between Llama3-8B-Instruct and Llama3.1-8B-Instruct in model features, and similar features are modeled at the similar layer.

### D.2 ANALYSIS OF REPRESENTATION EDITING METHOD

To investigate the effectiveness of the representation editing method in enhancing the model's ability to recognize conflict scenarios, we conducted a comparative analysis using linear probes. These probes were trained on the hidden states of the last layer of models that underwent fine-tuning and representation editing. Figure 9 illustrates our findings.

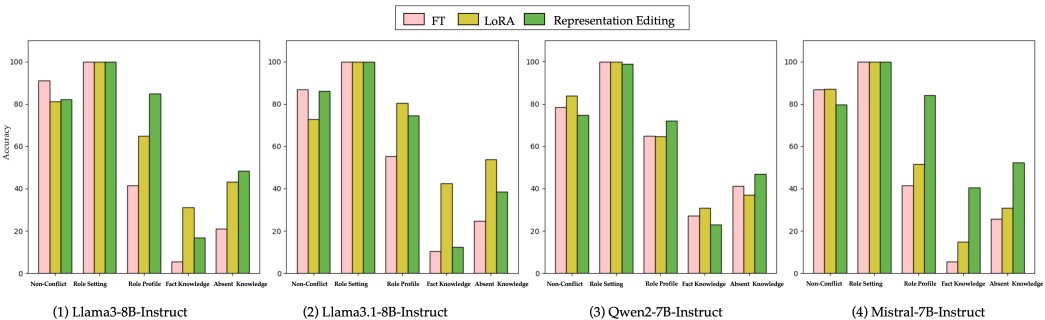

Figure 9: Accuracy of linear probes on the last layer for different query types.

The results reveal significant insights into how different methods affect the model's awareness across various scenarios:

**Well performance in contextual conflicts** In the two conflict types directly related to the character - "Role Setting" and "Role Profile" - the representation editing method demonstrated excellent performance across all models, typically outperforming or matching other methods.

**Improvement in parametric knowledge conflicts** In the two conflict types involving parametric knowledge - "Fact Knowledge" and "Absent Knowledge" - the representation editing method significantly outperformed FT and LoRA methods in most cases. This improvement is particularly evident in the Llama3-8B-Instruct and Mistral-7B-Instruct models.

### D.3 ANALYSIS OF REPRESENTATION VIA T-SNE

We also show the results of t-SNE visualization of the last layer representation of models, Llama3-8B-Instruct, Mistral-7B-Instruct, and qwen2-7B-Instruct, as shown in Figure 10.

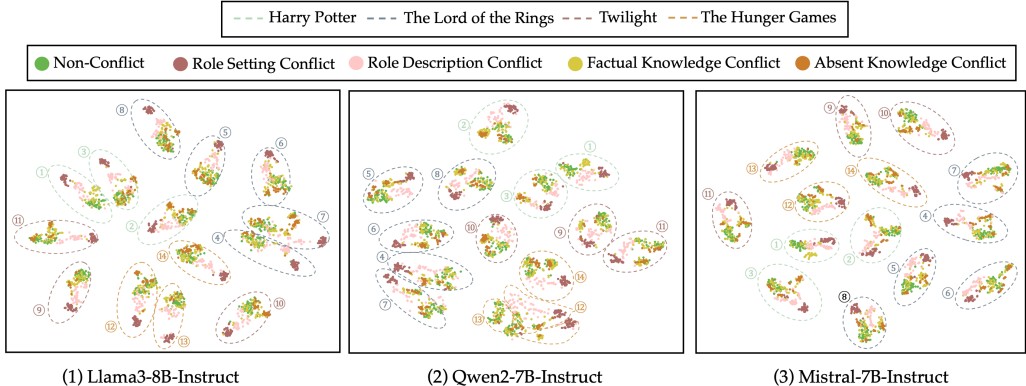

Figure 10: The results of visualizing the representations of the last layer using t-SNE.

From the analysis of additional t-SNE results, it is evident that the conclusions remain consistent across various models. These include distinct representation spaces for different roles, clustering of similar roles, clear separation of contextual knowledge conflict queries, and overlap of parametric knowledge conflict queries. This consistency reinforces the robustness of our findings across different model architectures.

### D.4 ANALYSIS OF COMPUTATION OVERHEAD

The representation editing method does not incur significant additional computational overhead. We analyze the computational overhead of our method mainly from two aspects: training overhead and inference overhead.

**1. Training Overhead:** As we have shown in Table 4 of our paper, our method does not involve any trainable parameters. Specifically, we only need to precompute and store the rejection vectors, which can then be simply added to the model's internal representations during practical applications. Therefore, compared to FT and LoRA, the computational overhead during the training phase of the representation editing method is nearly zero.

**2. Inference Overhead:** During inference, our method only requires a simple vector addition operation between the precomputed rejection vectors and the current internal representations of the model. This operation has a computational complexity similar to the adapter modules in LoRA. Since this operation is extremely lightweight, its impact on inference time and computational resources is almost negligible. Therefore, our method does not introduce significant additional overhead during the inference phase either.

