# OpenReview forum: "Tell Me What You Don't Know: Enhancing Refusal Capabilities of Role-Playing Agents via Representation Space Analysis and Editing"
_ICLR.cc/2025/Conference — Submitted to ICLR 2025_

### Official Review · Reviewer_GGuM · 2024-11-03

**Soundness:** 3
**Presentation:** 2
**Contribution:** 3
**Rating:** 5
**Confidence:** 3

**Summary:**

The paper discusses role-playing agents, which is of great interest. However, while the paper discuss the motivation for "enhancing of refusals", considering the refusal capabilities of state-of-the-art LLMs such as GPT4o, one wonders why refusal is such a problem, i.e., it seems to be possible to achieve it using RL or DPO, so what is the problem? While anyway there is merit in another method, the relation to existing refusals of LLMs (irrespective if its due to knowledge boundaries or ethical reasons) should be better discussed.
The paper also argues that while there exist a few approaches to handle the issue, a systematic evaluation is lacking. However, the paper fails to show convincingly why their approach is systematic and even scientific. That is, the refusal patterns come out of the blue, there is no detailed description of how they were derived so that they might be reproduced and potential gaps could be shown. While I understand that for CS conferences this is rarely done, it is still a shortcoming. As even in CS one would expect more elaboration and motivation. Thus, as a reader it is hard to assess the categories due to lack of depth.
That is, the paper might benefit from more focus, as it also aims to introduce a benchmark. However, here also the description is less than 2 pages, leaving many questions open. It is appreciated that data and code is open-sourced, but given the space constraints of conference papers, it seems close to impossible to aim for doing more than a benchmark (or method) within a paper.
On the positive side, the investigation of why there is a gap in handling different types of conflicting queries is interesting and maybe be worth expanding. Also the editing method based on Li et al. is interesting and more could be done in this direction.

Details:
* Abstract: Grammar issue:  we find that most RPAs behave significant performance gaps toward different conflict requests
Intro: Ideally, the example "Who murdered ..." in the intro is real not hypothetical. Otherwise it looks like a good example is hard to find...
* As a methodological shortcoming, the paper uses GPT4o in dataset construction and also evaluates GPT4o on it, claiming that it outperforms other models. While this might be technically correct, a NIPS paper from 2023/24 discussed at great length self-evaluation biases showing that model tend to better self-evaluate themselves. This has not been brought up in the paper.
* Conclusions: PRAs -> RPAs

**Strengths:**

see above

**Weaknesses:**

see above

**Questions:**

None, really.

---

> ### Author Response · Authors · 2024-11-19
> **Response to The importance of refusal capability is unclear.**
>
> **1. Importance of Refusal Capability:**
> While the latest LLMs like GPT-4 have demonstrated strong refusal capabilities, we find that there are still significant shortcomings in specific scenarios, particularly in role-playing. The uniqueness of RPAs lies in their need to handle various user requests appropriately while maintaining role consistency, including requests that may conflict with the role's settings or knowledge scope.
>
> **2. Why Refusal is Still a Problem:**
> - *Contextualized Refusal Needs:* In RPAs, refusal is not just a simple safety mechanism; it needs to be closely tied to the role's settings. The model must understand when to refuse and how to do so in a manner consistent with the role's style.
> - *Limitations of Existing Models:* Although models like GPT-4 have strong refusal capabilities in general scenarios, they still fall short in RPA scenarios, especially when queries conflict with the role's parametric knowledge.
>
> **3. Why Not Use DPO and RL**
> While RL and DPO have potential in enhancing specific model capabilities like refusal, we did not adopt these methods in this study for the following reasons:
>
> (1) **High Data Production Costs:**
> RL and DPO methods typically require a large amount of paired preference data, where the model needs to generate multiple candidate outputs for the same input, and these outputs are ranked by preference through human or automated means. Ensuring the model learns the correct preferences requires a substantial amount of high-quality annotated data. The process of ensuring data quality is complex, time-consuming, and labor-intensive.
>
> (2) **High Computational Resource Requirements:**
> RL and DPO methods require significant computational resources during training. The model may need to iterate repeatedly, and if using RL, it needs to evaluate and update strategies to maximize the reward function. For large language models, the training cost and time expenditure are substantial.
>
> Therefore, we opted for a representation editing method to address the dependency on preference data and training resources. The advantages of using the representation editing method include:
> (1) **Efficiency and Lightweight Nature:** Our representation editing method does not require large amounts of preference data or retraining the model. By intervening in the model's internal representations, we can enhance the model's refusal capability without altering its parameters.
> (2) **Low Data Requirements:** Compared to the large-scale preference data needed for RL and DPO, our method requires only a small amount of annotated data to achieve effective performance improvements, reducing data production costs.
>
> Overall, while existing LLMs possess some refusal capabilities, refusal remains a challenging issue in specific RPA scenarios. Our research aims to uncover these challenges and provide effective solutions to enhance the model's refusal capabilities in RPAs, ensuring that the model can interact with users safely and reliably while maintaining role consistency.

---

> ### Author Response · Authors · 2024-11-19
> **Response to Lack of discussion on the refusal mechanisms of existing LLMs.**
>
> Our research analyzed the refusal mechanisms of existing LLMs and found that their refusal capabilities are closely linked to internal representations. Based on this, we proposed a representation editing method that enhances the model's refusal capabilities by addressing its internal mechanisms.
>
> **1. Source of Refusal Mechanisms:**
> According to our analysis, the generation of refusal responses by a model is related to its internal mechanisms, specifically the presence of representations associated with refusal. When a query triggers the refusal mechanism, the representations related to refusal may become activated, leading the model to produce a refusal response. Through probe analysis and t-SNE visualization, we observed that the representations in refusal mode (queries that generate refusal responses) can be distinctly separated from those in non-refusal mode (queries that do not generate refusal responses), supporting this view to some extent.
>
> **2. Relationship Between Our Method and Existing Refusal Mechanisms:**
> Based on the understanding of the model's refusal mechanisms, we proposed the representation editing method, which aims to enhance refusal capabilities by directly addressing the model's internal mechanisms. Our representation editing method leverages and strengthens the model's existing refusal mechanisms without requiring additional training or changes to model parameters. This approach complements existing refusal mechanisms by providing a lightweight and effective method to enhance refusal capabilities from the perspective of internal representations.

---

> ### Author Response · Authors · 2024-11-19
> **Response to Why our benchmark and evaluation methods are systematic and scientific**
>
> In designing our benchmark and evaluation methods, we systematically constructed conflict query categories based on the sources of a model's knowledge and adopted a multi-dimensional evaluation strategy.
>
> 1. **Designing Conflict Query Categories from the Perspective of Knowledge Sources:**
>    Large language models primarily derive their knowledge from two sources: Contextual Knowledge and Parametric Knowledge. Based on this, we designed four conflict query categories, each targeting these two knowledge sources. This design approach covers the main types of knowledge conflicts that models might encounter, helping to deeply analyze the model's refusal capabilities across different knowledge sources and identify specific areas for improvement. Moreover, this classification method can be applied to other models and tasks, offering broad applicability.
>
> 2. **Comprehensive Evaluation of Model Capabilities from Three Main Perspectives:**
>    To thoroughly assess the model's performance, we evaluated it from three main dimensions: general conversational ability, role-playing ability, and refusal capability. These were further subdivided into: (1) General Conversational Ability: quality of response, consistency of response, factuality of response; (2) Role-Playing Ability: consistency with role background, style, personality, and abilities; (3) Refusal Capability: awareness of refusal and execution of refusal. This resulted in a detailed evaluation across nine specific dimensions. We noted that previous research often focused on only one or a few of these evaluation dimensions, such as assessing the personality of role-playing models or evaluating role-playing capabilities. We believe that only by combining these dimensions can we comprehensively evaluate and enhance a model's performance in complex interaction scenarios. Therefore, our research provides a more systematic and scientific evaluation framework.

---

> ### Author Response · Authors · 2024-11-19
> **Response to Concerns about bias in GPT evaluation**
>
> We want to clarify that the choice of GPT-4 as an evaluation tool is based on its demonstrated high consistency with human evaluations in existing research. As noted in [1] and [2], GPT-4 shows the highest alignment with human evaluation results when assessing text. This indicates that, despite potential self-evaluation biases, GPT-4 remains the most human-aligned automated evaluation tool currently available.
>
> Additionally, in our dataset construction process, we did not solely rely on model outputs; we also incorporated a human evaluation step. Specifically, after data generation, we randomly selected a portion of the samples for manual verification and assessment to ensure the quality and diversity of the data. Through this approach, we aim to minimize the impact of any biases that might arise from model self-evaluation.
>
> [1] Hackl, Veronika, et al. "Is GPT-4 a reliable rater? Evaluating consistency in GPT-4's text ratings." Frontiers in Education. Vol. 8. Frontiers Media SA, 2023.
>
> [2] Liu, Yang, et al. "G-eval: Nlg evaluation using gpt-4 with better human alignment." arXiv preprint arXiv:2303.16634 (2023).

---

> ### Author Response · Authors · 2024-11-19
> **Response to The cases are not realistic enough**
>
> We have provided a set of real failure cases. These cases consist of 10 refusal failures randomly sampled from the "Gandalf" role's absent conflict scenarios on GPT-4o. The data is stored in an [Excel file](https://anonymous.4open.science/r/Failure-Cases-of-Tell-Me-What-You-Don-t-Know-A3B4/Failure%20Cases.csv), which details each specific query and the corresponding response from GPT-4o. To ensure anonymity during the review process, we have stored these failure case data in an anonymous GitHub repository for reviewers to access.

---

> ### Author Response · Authors · 2024-11-19
> **Response to grammatical and spelling errors**
>
> Thank you for pointing out the grammatical and spelling errors. We will make the necessary corrections in the revised version.

---

> ### Comment · Reviewer_GGuM · 2024-11-21
> **Acknowledgement of author's response**
>
> The response is good, but after reading through the negative reviewer's comments critizing primarily the generalizability, I kind of agree with him though not to that extent, i.e. I keep my score

---

> > ### Author Response · Authors · 2024-11-22
> > **Response to concern about generalizability**
> >
> > Thank you for taking the time to review our response and for acknowledging our efforts to address the issues raised.
> > We would like to further clarify your concern about generalizability.
> > 1. **Applicability to Different Models:**
> >    Our experiments were conducted on multiple LLMs, including the Llama-3, Mistral, and Qwen series. Despite differences in training data, parameter sizes, and design, we observed consistent trends in their refusal capabilities and internal representation patterns across all models. This consistency indicates that our findings are not limited to specific models but are applicable to a range of LLMs.
> > 2. **Universality of Refusal Scenarios:**
> >    We designed our evaluation benchmark to cover various conflict scenarios, including contextual knowledge conflicts, parametric knowledge conflicts, and non-conflict queries. These scenarios represent the types of knowledge conflicts that RPAs may encounter in various applications such as virtual assistants, educational tools, and game NPCs. Our approach to designing conflicts based on the knowledge sources of large models can also be generalized to other scenarios.
> > 3. **General Representation Editing Method:**
> >    Although our research focuses on RPAs, the representation space analysis and the proposed representation editing method are not limited to role-playing. As demonstrated in [1], editing a model's representations can enhance its honesty, safety, fairness, and more, fully demonstrating that the representation editing method is also generalizable.
> > 4. **Laying a Foundation for Future Research:**
> >    The goal of our work is to provide a foundation for understanding the internal mechanisms that influence a model's refusal behavior. By identifying the existence of rejection regions and direct response regions within the model's representation space, we offer insights that can be utilized by future research to develop more robust and trustworthy AI models.
> >
> > Thank you again for your response. We hope that these additional explanations can alleviate your concerns about the generalizability of our work :)
> >
> > [1] Zou, Andy, et al. "Representation engineering: A top-down approach to ai transparency." arXiv preprint arXiv:2310.01405 (2023).

---

### Official Review · Reviewer_j72b · 2024-11-03

**Soundness:** 1
**Presentation:** 2
**Contribution:** 1
**Rating:** 3
**Confidence:** 4

**Summary:**

The paper considers the problem of “refusal capabilities” of LLM agents. The authors in particular identify “rejection regions” using a t-SNME approach. They claim that they are able to distinguish different types of “rejections" based on this type of analysis

**Strengths:**

- In general, the questions proposed by the authors are interesting and worth investigating for their potential implications.

**Weaknesses:**

- The authors claim that they are able to demonstrate the existence of rejection regions and direct response regions. However, this does not seem to be the case in general. In fact, it seems to me that the actual definitions of these two concepts are not “rigorous” per se.
- The reviewer understands that it is difficult to run experiments that are “generalizable,” but it seems to me that the actual analysis and discussion proposed in the paper tend to consider the experimental results presented as universal, whereas they are probably very specific to the context taken into consideration. The scenarios defined by the authors are somehow very specific. It is difficult to understand how these results will generalize in practice.
- The actual identification of these regions is performed through t-SNE, but very limited details are provided to the reader. It appears that the reproducibility of these experiments is rather limited in general.
- The authors use GPT-4 for the data generation and synthesis. The actual impact of this choice is difficult to evaluate. In fact, it might be the case that these questions/answers are already “biased” in a sense: the reviewer wonders if the results presented in the paper might just be an artifact linked to the use of GPT-4.
- In general, the reviewer understands that it might be very difficult to generate datasets in this area, but the results presented by the authors might be really dependent on the actual generation procedure used by the authors. This aspect should be at least discussed in the paper.
- The description of the procedure for linear probing (see also Appendix C.2) is not described in sufficient detail. It is very difficult to judge the results without additional details.

**Questions:**

- What is the definition of “refusal capability” that you are actually considering in this paper? It seems to me that this concept should be clarified. In fact, the reviewer has some intuition of the problem (from the examples/description), but this concept is not formally introduced in the paper, in my opinion.
- Can you please describe the t-SNE process in detail, including the choice of parameters?
- What is the impact of using GPT-4 for data generation?
- Can you please describe the procedure used for linear probing in more detail (see Appendix C.2)?
- In Section 5.1, the authors say: “In contrast, the lower accuracy of the probes for parametric knowledge conflicts indicates that models struggle to internally differentiate these conflicts from non-conflict queries.” What do you mean by “internally differentiate” here?
- The reviewer struggles to understand why t-SNE is a good choice in this case. Can you please justify your selection of the algorithm?
- In Section 5.2, the description of the method used for separating the different regions is difficult to understand. Can you please provide more details?
- The reviewer was not able to understand the “representation activation method” presented in Section 6.1. Can you please clarify it? There are also other concepts that are loosely defined, such as “rejection direction.” A formal definition of this term would be very useful.
- In Section 6.2, why do you call fine-tuning and LORA baselines in this context?
- Figures 3 and 10: The areas are composed of different types of conflicts. What does this mean in practice?
- Figure 10: The authors say: “This consistency reinforces the robustness of our findings across different model architectures.” Why do you say that “consistency reinforces robustness of our findings across different model architectures”? To which consistency are you referring here? This is rather unclear. Referring to Figure 10, the reviewer wonders if the results we observe in this figure are more related to the topics of the sentences themselves (and/or the presence of some keywords).
- What are the general implications of this work besides the specific examples considered by the authors? How can we generalize the results presented in this work?

**Details Of Ethics Concerns:**

The reviewer would like to raise some concern about the potential double use of these techniques.

For example, in non-democratic and authoritarian regimes, they can be used to limit the freedom of speech by avoiding potential difficult questions about historical events that a regime would like to "remove from history", questions about political and socio-economic issues, etc.

---

> ### Author Response · Authors · 2024-11-19
> **Response to W1: Definition of Rejection Regions and Direct Response Regions**
>
> Regarding your concern about the definitions of "rejection regions" and "direct response regions" not being rigorous, we would like to offer clarification.
>
> Firstly, in our paper, we did not claim to **demonstrate** the existence of these two regions; rather, we aimed to **reveal** the distinct response patterns of the model to different types of queries through extensive experimental observations.
>
> Our preliminary definitions of these concepts are based on empirical observations:
>
> - **Rejection Regions**: When the similarity between the input query's representation vector $\\mathbf{h}^l$ and the rejection direction vector $\\mathbf{d}'^l$ exceeds a certain threshold $\\theta$, i.e., $\\text{sim}(\\mathbf{h}^l, \\mathbf{d}'^l) \\geq \\theta$, the model is more inclined to trigger the refusal mechanism and decline to answer the query.
>
> - **Direct Response Regions**: When the similarity is below the threshold $\\theta$, i.e., $\\text{sim}(\\mathbf{h}^l, \\mathbf{d}'^l) < \\theta$, the model tends to generate a direct response to the query.
>
> The similarity $\\text{sim}(\\mathbf{h}^l, \\mathbf{d}'^l)$ can be calculated using cosine similarity:
>
> $$
> \\text{sim}(\\mathbf{h}^l, \\mathbf{d}'^l) = \\frac{\\mathbf{h}^l \\cdot \\mathbf{d}'^l}{\\|\\mathbf{h}^l\\| \\|\\mathbf{d}'^l\\|}
> $$
>
> We acknowledge that due to the high dimensionality and complex nonlinear characteristics of large language models' representation spaces, providing strict mathematical definitions and proofs is challenging. Therefore, our definitions are more intuitive descriptions based on experimental phenomena. In our research, we observed similar patterns across various large language models (such as Llama-3, Mistral, Qwen) and different conflict scenarios. Despite differences in architectures and training data, these models consistently exhibited differentiated treatment between conflicting and non-conflicting queries in the representation space. This consistency further supports our assertion regarding the existence of rejection regions and direct response regions.

---

> > ### Comment · Reviewer_j72b · 2024-11-22
> >
> > This was my concern: the fact that the authors include several experiments does not guarantee the fact that this is the case in general. This seems quite problematic to me since it is an essential point of the the proposed approach.

---

> ### Author Response · Authors · 2024-11-19
> **Response to W2: Refusal scenarios lack generalizability.**
>
> Regarding your concern that our analysis and discussion may present the experimental results as universally applicable, while they might be specific to certain contexts, we would like to provide a detailed response.
>
> **1. Importance of Refusal Capabilities:**
> We believe that training models to have refusal capabilities is crucial for developing reliable AI assistants. The ability of a model to clearly recognize its knowledge boundaries and appropriately refuse to answer questions beyond its scope or capability not only enhances the model's reliability but also prevents the provision of incorrect or misleading answers. This refusal capability is significant across a wide range of applications, not limited to simple question-answering systems. In broader natural language processing tasks, models are often assigned specific roles to complete tasks. For example:
> - **Mathematical Problem Solving**: A model acts as a mathematician to solve complex mathematical problems. When faced with problems beyond its understanding, the model can appropriately refuse, avoiding incorrect solutions.
> - **Medical Diagnosis**: A model acts as a doctor to provide advice to patients. If the model can refuse to answer in uncertain situations and suggest seeking professional help, it can prevent risks associated with incorrect diagnoses.
> - **Role-Playing Chat and Game NPCs**: In dialogue systems and games, models play specific roles to interact with users. Characters with refusal capabilities are more realistic and can enhance user experience. For instance, an NPC in a game will not answer questions beyond its set scope, increasing the game's immersion.
>
> Therefore, researching and enhancing a model's refusal capabilities has broad applicability for building reliable and intelligent AI systems. This is not only relevant to the specific scenarios discussed in our paper but also important for various applications where models need to understand their knowledge boundaries.
>
> **2. Universality of Scenario Design:**
> Our scenario design is based on the knowledge sources of large language models, namely contextual knowledge (immediate information provided through input) and parametric knowledge (intrinsic knowledge obtained through pre-training). We believe that conflicts between these two knowledge sources are common issues that models encounter in practical applications. Designing experimental scenarios around these knowledge conflicts helps comprehensively evaluate and enhance the model's refusal capabilities.
>
>
> **3. Generalizability of Experimental Results:**
> We acknowledge that the experiments were conducted in specific scenarios and role settings. However, the scenarios we chose are intended to represent typical issues that models might encounter in practical applications, such as knowledge conflicts in role-playing and the model's awareness of its knowledge boundaries. These are common challenges across various application scenarios. We understand your concern that the experimental results might be limited by specific scenarios. To verify the broader applicability of our findings, we conducted experiments across multiple models (such as Llama-3, Qwen2, etc.) and different types of conflict scenarios, and the results showed consistent trends. This indicates that our approach has a certain degree of generalizability across different models.
>
> We understand your concerns about the generalizability of the experimental results. However, we believe that while our experimental scenarios involve specific roles and contexts, the underlying principles and methods have universal applicability. The refusal capability of models is key to building reliable AI systems, and the knowledge conflicts and refusal mechanisms we explore are significant in various applications.

---

> ### Author Response · Authors · 2024-11-19
> **Response to W4&Q3: Concerns about Synthetic Data**
>
> **1. Rationale for Using GPT-4 for Data Generation:**
>
> In current natural language processing research, utilizing large pre-trained models like GPT-4 for data synthesis has become a common and effective approach [1]. Numerous studies have demonstrated [2][3] that data generated by GPT can enhance model performance and provide high-quality data support, especially in the absence of large-scale annotated datasets. For instance, researchers have successfully used GPT-generated data to train and evaluate models, achieving significant results.
>
> **2. Design and Quality Assurance of the Data Generation Process:**
>
> When using GPT-4 for data generation, we implemented a rigorous process design and quality control measures, as detailed in Section 3.2 of our paper, to ensure the validity and reliability of the data:
>
> - **Diversity and Coverage**: We carefully designed the prompts for data generation to ensure that the generated questions and answers cover a wide range of scenarios and types, avoiding data uniformity and bias.
>
> - **Data Quality Filtering**: After data generation, we incorporated a data quality filtering step, using both automated and manual methods to remove low-quality or non-compliant data. This includes eliminating duplicates, ensuring the reasonableness of questions, and the accuracy of answers.
>
> - **Manual Verification**: To further ensure data quality, we conducted random sampling and manual verification of the generated data, assessing its fluency, relevance, and accuracy.
>
> **3. On Potential Bias and Impact:**
>
> Our main conclusions are based on multiple models and diverse datasets, and we observed consistent results across different experimental settings. This indicates that our findings are not merely artifacts of GPT-4 data generation but have a certain degree of generalizability.
>
> [1] Qin, Yulei, et al. "Unleashing the power of data tsunami: A comprehensive survey on data assessment and selection for instruction tuning of language models." arXiv preprint arXiv:2408.02085 (2024).
>
> [2] Ding, Ning, et al. "Enhancing chat language models by scaling high-quality instructional conversations." arXiv preprint arXiv:2305.14233 (2023).
>
> [3] Taori, Rohan, et al. "Alpaca: A strong, replicable instruction-following model." Stanford Center for Research on Foundation Models. https://crfm. stanford. edu/2023/03/13/alpaca. html 3.6 (2023): 7.

---

> ### Author Response · Authors · 2024-11-19
> **Response to W5: Question About The Details of Data Generation.**
>
> In Section 3.2 "Data Construction" of our paper, we have provided a thorough description of how we built our dataset. Specifically:
>
> - **Multi-Source Data Collection**: We extended the existing TIMECHARA dataset to construct our RoleRef dataset by integrating real dialogue data, publicly available data resources, and synthetic data generated using large language models like GPT-4. This approach ensured that our dataset was diverse and comprehensive.
>
> - **Designing Diverse Refusal Scenarios**: To comprehensively evaluate the refusal capabilities of models, we meticulously designed various conflict scenarios. These include role setting conflicts, role description conflicts, factual knowledge conflicts, and absent knowledge conflicts. Each scenario is representative and diverse, targeting different types of knowledge conflicts that models might encounter in practice.
>
> - **Data Generation Strategies**: We combined predefined templates with automated tools to generate a large volume of high-quality queries. For each generated query, we provided corresponding reference answers and explanations to ensure the accuracy and reliability of the data.
>
> - **Manual Verification**: To further ensure data quality, we performed random sampling and manual verification of the generated data. This process involved assessing the fluency, relevance, and accuracy of the queries and responses.
>
> **2. Discussion on the Dependency of Results:**
>
> We understand your concern that our experimental results might depend heavily on the specific data generation process. To address this, we have implemented the following measures to ensure the robustness and generalizability of our findings:
>
> - **Multi-Model Validation**: We conducted experiments across multiple models with different architectures and scales, including the Llama-3 and Qwen2 series. These models vary in training data, parameter sizes, and design. Despite these differences, our experiments demonstrated consistent trends across all models. This consistency suggests that our findings are not specific to any single model, indicating a certain level of universality and robustness.
> - **Multi-Scenario Testing**: Our experiments encompass a wide range of conflict and non-conflict query types. By evaluating model performance across different scenarios, we verified the stability of our results and minimized the potential impact of the data generation process on our conclusions.
>
> These efforts collectively ensure that our results are not artifacts of a particular data generation method but are indicative of broader trends in model behavior. By rigorously designing our data generation process and thoroughly validating our findings across multiple models and scenarios, we have strengthened the generalizability and practicality of our research.

---

> ### Author Response · Authors · 2024-11-19
> **Response to Q1: What is the definition of the ability to refuse to answer?**
>
> In our study, "refusal capability" refers to **a model's ability to provide correct answers to questions within its knowledge scope while appropriately refusing to answer questions that fall outside of this scope.**
>
> We define the model's knowledge scope from two perspectives:
> 1. **Contextual Knowledge**: This refers to the knowledge explicitly provided in the context during interactions. The model should use this immediately available knowledge to answer relevant questions.
> 2. **Parametric Knowledge**: This encompasses the knowledge learned and internalized in the model's parameters through the pre-training process. It represents the long-term knowledge reserves acquired from training on extensive corpora.
>
> The refusal capability of a model is specifically manifested in three major aspects:
> 1. **Conflict Recognition Ability**: The ability to identify queries that conflict with role contextual knowledge and role parametric knowledge.
> 2. **Refusal Response Ability**:
>    - **Providing Clear Refusal Responses**: The model should clearly express its inability to answer the question.
>    - **Explaining the Reason for Refusal**: Appropriately explaining why it cannot answer helps the user understand.
>    - **Maintaining Consistency with Role Characteristics**: When refusing, the model should maintain the language style and personality traits of the role.
>    - **Offering Alternative Information or Clarifications When Appropriate**: If possible, the model can provide relevant suggestions or request further clarification from the user.
> 3. **Refusal Accuracy**:
>    - **Avoiding Over-Refusal**: The model should not incorrectly refuse normal, non-conflict queries.
>    - **Avoiding Missed Refusals**: For conflict queries, the model should accurately identify and refuse them, rather than incorrectly providing an answer.
>
> Through these three core dimensions, we comprehensively define the "refusal capability" of a model. This not only requires the model to correctly identify when a refusal is necessary but also to refuse in a manner consistent with the role's characteristics, ensuring coherence in interaction and a positive user experience.

---

> ### Author Response · Authors · 2024-11-19
> **Response to Q5: What is internally differentiate mean?**
>
> Previous research has shown that when LLMs process inputs, their internal representations reflect the model's grasp of the knowledge within the query. For example, [1] and [2] have pointed out that a model's internal representations can reveal its ability to distinguish between known and unknown information.
>
> **"Internal distinction" refers to the model's ability to recognize and differentiate different types of queries within its internal representations.** Our experimental results indicate that the model lacks this internal distinction capability when dealing with parametric knowledge conflict queries, leading to an inability to appropriately refuse to answer such queries. In contrast, for contextual knowledge conflict queries, the model can effectively distinguish internally and thus adopt appropriate refusal strategies.
>
> [1] Azaria, Amos, and Tom Mitchell. "The internal state of an LLM knows when it's lying." arXiv preprint arXiv:2304.13734 (2023).
>
> [2] Ji, Ziwei, et al. "Llm internal states reveal hallucination risk faced with a query." arXiv preprint arXiv:2407.03282 (2024).

---

> > ### Comment · Reviewer_j72b · 2024-11-22
> >
> > This does not appear the meaning of your analysis?

---

> ### Author Response · Authors · 2024-11-19
> **Response to Q8: Formal Definition**
>
> We recognize the need to provide a more detailed explanation of the representation editing method in Section 6.1, including formal definitions of key concepts. To help you better understand, we will detail the three steps of the method and provide formal definitions for each step.
>
> **Detailed Process of the Representation Editing Method:**
>
> ---
>
> **Step 1: Collecting Activation**
>
> For each role, we construct a set of **conflict queries** and **non-conflict queries**, represented as:
>
> - Conflict query set: $\\{ q_{\\text{conflict}}^i \\}_{i=1}^N$
> - Non-conflict query set: $\\{ q_{\\text{non-conflict}}^i \\}_{i=1}^N$
>
> For each query $q$, we obtain the model's hidden state representation at **each layer**, denoted as:
>
> - Conflict query representation at layer $l$: $\\mathbf{h}_{\\text{conflict}}^{i,l}$
> - Non-conflict query representation at layer $l$: $\\mathbf{h}_{\\text{non-conflict}}^{i,l}$
>
> where $l = 1, 2, \\dots, L$, and $L$ is the number of layers in the model.
>
> ---
>
> **Step 2: Identifying the Rejection Direction**
>
> In this step, we calculate the representation differences between conflict and non-conflict queries at each layer to capture the features associated with the model's refusal behavior.
>
> For each layer $l$, compute the representation difference vector for the $i$-th query pair:
>
> $$
> \\Delta \\mathbf{h}^{i,l} = \\mathbf{h}_{\\text{conflict}}^{i,l} - \\mathbf{h}_{\\text{non-conflict}}^{i,l}
> $$
>
> Then, calculate the average of all difference vectors to obtain the **rejection direction** $\\mathbf{d}^l$ at layer $l$:
>
> $$
> \\mathbf{d}^l = \\frac{1}{N} \\sum_{i=1}^N \\Delta \\mathbf{h}^{i,l}
> $$
>
> To filter out noise and retain features highly related to refusal behavior, we compute the variance for each dimension of the difference vectors. Let $\\sigma_{l,j}^2$ be the variance of the $j$-th dimension at layer $l$. We zero out dimensions with variance above a threshold $\\tau$, resulting in the adjusted rejection direction $\\mathbf{d}'^l$:
>
> $$
> \\mathbf{d}_{j}'^l = \\begin{cases}
> \\mathbf{d}_{j}^l, & \\text{if } \\sigma_{l,j}^2 \\leq \\tau \\\\
> 0, & \\text{if } \\sigma_{l,j}^2 > \\tau
> \\end{cases}
> $$
>
> ---
>
> **Step 3: Steering Activation**
>
> With the rejection direction for each layer, we intervene in the model's internal representations when processing new queries.
>
> For a new query $q$, obtain its hidden state representation at layer $l$, $\\mathbf{h}^l$.
>
> Calculate the similarity between $\\mathbf{h}^l$ and the rejection direction $\\mathbf{d}'^l$, for example, using cosine similarity:
>
> $$
> \\text{sim}(\\mathbf{h}^l, \\mathbf{d}'^l) = \\frac{\\mathbf{h}^l \\cdot \\mathbf{d}'^l}{\\|\\mathbf{h}^l\\| \\|\\mathbf{d}'^l\\|}
> $$
>
> If the similarity exceeds a set threshold $\\theta$, the query at layer $l$ may require intervention. We add the rejection direction to the original representation proportionally by $\\lambda$:
>
> $$
> \\mathbf{h}^{l} \\leftarrow \\mathbf{h}^{l} + \\lambda \\mathbf{d}'^l
> $$
>
> By adjusting the representations at each layer, we gradually guide the model to be more inclined to refuse to answer conflict queries.
>
> ---
>
> **Formal Definitions of Concepts:**
>
> - **Rejection Direction $\\mathbf{d}^l$**: At layer $l$, it is the average representation difference when the model processes conflict versus non-conflict queries, capturing features of the model's refusal behavior.
>
> - **Adjusted Rejection Direction $\\mathbf{d}'^l$**: Obtained by filtering $\\mathbf{d}^l$ based on variance, retaining features highly related to refusal behavior.
>
> Through this method, we can enhance the model's ability to refuse conflict queries by leveraging its representations at each layer, without altering the model's parameters.

---

> > ### Comment · Reviewer_j72b · 2024-11-22
> >
> > These definitions are rather useful.

---

> ### Author Response · Authors · 2024-11-19
> **Response to Q9: Baseline selection**
>
> We consider Fine-tuning and LoRA as baseline methods for the following reasons:
>
> 1. **Shared Research Objective**: Our proposed Representation Editing method aims to enhance the model's ability in refusal scenarios while maintaining its original performance as much as possible. Fine-tuning and LoRA are also commonly used methods aimed at improving a model's refusal capabilities. Using them as baselines allows for a fair comparison under the same objectives, providing an objective assessment of the effectiveness of our method.
>
> 2. **Representativeness and Prevalence**: Fine-tuning and LoRA are widely adopted in research to enhance the performance of large language models. Many previous studies have used these methods to improve a model's refusal capabilities. Choosing them as baselines provides representativeness and reference value, helping to demonstrate the advantages of our method over existing commonly used methods.
>
> Therefore, in Section 6.2, we selected Fine-tuning and LoRA as baseline methods to objectively and comprehensively evaluate the effectiveness and advantages of our proposed Representation Editing method. By comparing with these commonly used methods, we aim to demonstrate that our method can effectively maintain or even enhance the model's original performance while improving its refusal capabilities.

---

> ### Author Response · Authors · 2024-11-19
> **Response to Q11: Explain the meaning of consistency.**
>
> 1. **Meaning of "Consistency":**
>
> **"Consistency" refers to the similar characteristics observed in the distribution patterns of internal representations across different model architectures (e.g., Llama3-8B-Instruct, Mistral-7B-Instruct, Qwen2-7B-Instruct).** Specifically, this consistency is reflected in: (1) Different roles having distinct representation spaces; (2) Roles from the same series (e.g., the same novel or story) being closer in the representation space; (3) Separation of representations for contextual knowledge conflict queries from non-conflict queries; (4) Overlap of representations for parametric knowledge conflict queries with non-conflict queries.
>
> 2. **Why "Consistency" Enhances Robustness of Results:**
>
> We assert that "this consistency enhances the robustness of our findings across different model architectures" because:
> - **Similar Performance Across Models**: Different model architectures exhibit similar internal representation distributions when processing the same types of queries. This indicates that our findings are not specific to a single model but have general applicability.
> - **Reliability of Results**: If different models show consistency in internal representation patterns, our analysis and conclusions about model behavior become more convincing and reliable.
> - **Commonality in Model Behavior**: Consistency reflects that different models may follow similar mechanisms or strategies when processing information, helping us understand the universal characteristics of models.
>
> 3. **Whether Results Are Related to Sentence Topics or Keywords:**
>
> We understand your concern that these results might primarily be due to the presence of specific topics or keywords in the queries. To address this, we took the following measures during data generation to ensure query diversity and avoid bias towards specific topics or keywords:
>
> - **Increasing Query Diversity**: When constructing queries, we used various question forms, including "when," "how," "where," etc., to avoid relying solely on specific sentence structures or keywords. This made the queries richer in syntactic structure and vocabulary, reducing the influence of specific keywords on the model.
> - **Ensuring Topic Diversity Based on Original Novel Content**: Our queries were generated based on the original novel texts, covering different events, character relationships, and plots in the novels. This ensured topic diversity and prevented the queries from being overly concentrated on specific topics, requiring the model to handle a variety of semantic content.
>
> Therefore, when we refer to **"consistency," we mean the similarity in internal representation distribution patterns across different models.** This consistency enhances the robustness of our findings because it indicates that our conclusions are applicable across different model architectures, demonstrating generality.

---

> ### Author Response · Authors · 2024-11-19
> **Response to Q12: General Implications**
>
> **1. Importance of the Research Problem (Refusal Capability):**
>
> The core of our research is to enhance the refusal capability of LLMs in role-playing agents (RPAs). This capability is crucial for developing trustworthy and reliable AI systems. Models with refusal capabilities can identify and appropriately refuse to respond to requests that are beyond their knowledge scope, conflict with their role settings, or contain inappropriate content. This not only improves the safety and reliability of the models but also enhances user experience by preventing the dissemination of incorrect information.
>
> **2. Our Contributions and Their General Significance:**
>
> While our experiments primarily focus on role-playing scenarios, our findings and methods have broad applicability and offer important insights for the wider fields of AI and natural language processing.
>
> (1) **Evaluation of Current Models' Refusal Capabilities:**
>
> We assessed the performance of existing mainstream large language models in handling different types of conflict queries, revealing differences in how models handle contextual knowledge conflicts versus parametric knowledge conflicts. This evaluation not only provides a baseline for understanding the current state but also highlights potential risks in model safety and reliability.
>
> *General Significance*: This evaluation method and scenario design can be extended to other tasks and fields, helping developers identify performance deficiencies in models under different contexts and make targeted improvements.
>
> (2) **Exploration of Why Models Perform Differently on Various Queries:**
>
> By analyzing the internal representations of models, we discovered differences in the internal mechanisms when handling different types of conflict queries, particularly the lack of internal differentiation capability in parametric knowledge conflicts. This finding reveals the connection between a model's internal representations and its behavior.
>
> *General Significance*: This deep understanding of internal mechanisms not only helps explain model behavior but also provides new perspectives for other researchers to analyze and improve model performance across different tasks.
>
> (3) **Proposal of a Representation Editing Method Based on Exploration Results:**
>
> We proposed a Representation Editing method that enhances a model's refusal capability without altering its parameters. This method adjusts the model's internal representations to make it more inclined to refuse conflict queries.
>
> *General Significance*: This method is a lightweight and versatile technique applicable to various models and tasks. It offers a new approach to improving model safety and reliability without affecting its original capabilities.
>
> **3. Generalization and Application of Results:**
>
> (1) **Applicable to Various Application Scenarios:**
>
> Refusal capability is necessary for any AI system that requires interaction with users. For example:
>
> - **Virtual Assistants and Dialogue Systems**: Improve the model's ability to handle inappropriate or out-of-scope questions, avoiding misleading users.
> - **Content Moderation and Safety Filtering**: Enable models to identify and refuse to generate harmful or policy-violating content.
> - **Professional Field Applications**: In fields like healthcare, law, and education, ensure that models do not provide incorrect information when encountering knowledge gaps.
>
> (2) **Implications for Model Training and Improvement:**
>
> Our research emphasizes the importance of focusing on internal representations, encouraging consideration of how models can better understand and manage their knowledge boundaries during training and design. This aids in developing more controllable and trustworthy AI systems.
>
> (3) **Promotion of AI Safety and Ethics:**
>
> Enhancing a model's refusal capability helps prevent the spread of incorrect information and protects users from potential misinformation. This aligns with AI safety and ethical principles, contributing positively to industry development.
>
> **Conclusion:**
>
> In summary, while our research uses role-playing agents as an example, its methods and findings have broad applicability and general significance. We believe that by evaluating and enhancing models' refusal capabilities, understanding performance differences across queries, and proposing general improvement methods, our work provides valuable references for building safer, more reliable, and trustworthy AI systems.

---

> ### Author Response · Authors · 2024-11-19
> **Response to W3&Q2&Q6&Q7&Q10: Detail of t-SNE**
>
> **Q7: Method for Separating Regions**
>
> We use t-SNE to visualize and analyze the model's representation space to identify the distribution of different types of queries (such as conflict and non-conflict queries) within the representation space.
>
> **Q6: Reasons for Choosing t-SNE**
>
> 1. **Need for Dimensionality Reduction:**
>    In our study, the model's representations are typically high-dimensional, making direct analysis and visualization challenging. To better understand and analyze these high-dimensional data, we need to reduce them to a more manageable space.
>
> 2. **Reasons for Choosing t-SNE:**
>    t-SNE is a nonlinear dimensionality reduction technique particularly suited for visualizing high-dimensional data. It effectively preserves local structures, ensuring that similar data points remain close in the low-dimensional space. This is crucial for identifying patterns and structures (such as rejection and direct response regions) in our model's representations.
>
> t-SNE has been widely used as a tool for representation analysis. In many studies, t-SNE has proven to be an effective tool for analyzing and visualizing high-dimensional data [1][2][3].
>
> **Q10: Practical Significance of t-SNE**
>
> In Figures 3 and 10, we show the distribution of different types of queries in the representation space. These regions represent the internal representation states of the model when processing different types of queries. By identifying these regions, we can better understand the model's decision-making process and further optimize its refusal capabilities. As introduced in Section 6.1, our representation editing method can adjust the model's representations to ensure more conflict queries fall into the rejection region, thereby enhancing the model's refusal capability.
>
> **W3&Q2: Detailed Explanation of the t-SNE Visualization Process**
>
> As mentioned in Section 5.2, we use t-SNE to reduce the dimensionality of and visualize the hidden states of the model's last layer to analyze the distribution of different types of queries in the model's internal representation space. The main steps of t-SNE can be divided into:
>
> 1. Data Collection
> 2. t-SNE Dimensionality Reduction
> 3. Visualization
>
> Detailed steps are as follows:
>
> 1. **Data Collection:**
>    - **Model Input:** We format the queries according to the prompt template shown in Figure 5 as input to the model.
>    - **Extracting Representations:** For each query type (non-conflict queries, contextual knowledge conflict queries, parametric knowledge conflict queries, etc.), we extract the hidden states of the last token from the model as the embedding for that query. To eliminate the influence of the token's inherent meaning, we standardize the last token to an end-of-text symbol, such as `<|eot_id|>` for `Llama-3.1-8B-Instruct`.
>    - **Sample Size:** For each query type, we select 50 samples from the test set for t-SNE visualization.
>
> 2. **t-SNE Parameter Settings:**
>    - **Algorithm Implementation:** We use the `TSNE` function from Python's scikit-learn library. For parameter selection, we use its default settings.
>      ```python
>      from sklearn.manifold import TSNE
>      tsne = TSNE(n_components=2, random_state=42)
>      data_2d = tsne.fit_transform(data)
>      ```
>
> 3. **Ensuring Reproducibility:** To further enhance reproducibility, we plan to provide a complete code example for t-SNE visualization in our code repository, including the full process of data extraction, dimensionality reduction, and plotting.
>
>
> [1] Zou, Andy, et al. "Representation engineering: A top-down approach to ai transparency." arXiv preprint arXiv:2310.01405 (2023).
>
> [2] Ji, Ziwei, et al. "Llm internal states reveal hallucination risk faced with a query." arXiv preprint arXiv:2407.03282 (2024).
>
> [3] Li, Tianlong, Xiaoqing Zheng, and Xuanjing Huang. "Open the Pandora's Box of LLMs: Jailbreaking LLMs through Representation Engineering." arXiv preprint arXiv:2401.06824 (2024).

---

> ### Author Response · Authors · 2024-11-19
> **Response to W6&Q4:Detail of Probe**
>
> 1. **Data Preparation:**
>    - **Hidden Representation Extraction:** For each query, we first use the prompt shown in Figure 5 as input to the model. During the model's forward pass, we extract the hidden states from a specified layer (e.g., the penultimate layer) to use as feature vectors.
>    - **Dataset Construction:** We collect the corresponding hidden representations for different types of queries (e.g., non-conflict queries, contextual conflict queries, parametric knowledge conflict queries). For each type of query, we use 200 samples for training and 50 samples for testing.
>    - **Label Assignment:** For binary classification, we assign a label of 1 to non-conflict query samples and a label of 0 to conflict query samples.
>
> 2. **Model Definition:**
>    - **Linear Probe Structure:** We use a simple fully connected neural network with one hidden layer (512 nodes) and an output layer with a Sigmoid activation function. This setup is used to probe whether the model perceives a query as conflicting with its knowledge.
>
> 3. **Training Process:**
>    - **Loss Function:** We use the Mean Squared Error Loss (MSELoss) to optimize the model parameters.
>    - **Optimizer and Hyperparameters:** We use the Adam optimizer with a learning rate of 5e-5, a batch size of 512, and train for 10 epochs.
>    - **Training Strategy:** The model is trained on the training set, and at the end of each epoch, its performance is evaluated on the validation set. The model parameters with the highest validation accuracy are saved.
>
> 4. **Result Evaluation:**
>    - **Evaluation Metrics:** We calculate the prediction accuracy for each query type on the test set to assess the linear probe's performance in distinguishing between different types of queries.
>    - **Experiment Reproducibility:** To ensure the reliability of the results, we set a fixed random seed and conduct experiments on data from multiple roles, calculating the average performance.

---

> ### Author Response · Authors · 2024-11-19
> **Response to Ethical Concerns**
>
> **1. Emphasis on Ethical Concerns:**
>
> We fully understand and agree with your concerns about the potential misuse of technology. **Any new technology and method can be used for both positive and negative purposes.** As researchers, we have a responsibility to consider the potential impacts of the technologies we develop and strive to ensure they are used to promote societal well-being.
>
> **2. Intent and Goals of the Technology:**
>
> **The technology we propose aims to enhance the safety and reliability of large language models.** Specifically, we want models to be able to identify and appropriately refuse to respond to requests that are beyond their knowledge scope, conflict with their role settings, or contain inappropriate content. This helps prevent the generation of incorrect, misleading, or harmful information, protecting users from potential negative impacts.
>
> **3. On the Limitation of Free Speech:**
>
> We also recognize that freedom of speech is an important societal value. However, we believe that appropriately limiting certain harmful content is necessary to protect users and society. For example, the dissemination of content related to violence, pornography, and racial discrimination can have negative impacts on society and even threaten public safety and moral standards. Therefore, in these cases, it is reasonable and necessary to restrict the spread of such content.
>
> In conclusion, we believe that addressing the issue of technology misuse requires the joint efforts of researchers, policymakers, industry, and society at large. We are willing to actively participate in related discussions and collaborations to promote the ethical and responsible application of AI technology.

---

> > ### Comment · Reviewer_j72b · 2024-11-22
> > **Acknowledgment of the Rebuttal**
> >
> > I read all the authors' comments. The rebuttals are very extensive in terms of length, but they do not actually clarify some key concerns I expressed in my review.
> >
> > I am rather negative about this paper; I have several reservations about its contents.

---

> > > ### Author Response · Authors · 2024-11-27
> > >
> > > Thank for your continued attention and feedback. We understand your concerns about the length of our responses. The extensive responses were intended to address all questions you raised comprehensively. Now, we would like to focus on addressing your key concerns:
> > >
> > > 1. Regarding the generalizability of experimental results:
> > > We understand your concern - good performance across multiple experiments does not fully guarantee the generalizability of the method and the scenarios. We would like to offer the following clarifications:
> > >
> > > - Comprehensiveness of experimental design: We conducted tests across different model architectures (Llama-3, Mistral, Qwen, etc.), and the results showed consistent trends.
> > > - Cost-benefit trade-off: While extending to more scenarios could further validate the method's generalizability, this would incur significant costs in data generation and verification. We believe the current experimental scope reasonably demonstrates both the effectiveness of our method and the soundness of our scenario design.
> > > - Application potential: As Reviewer `oci5` noted, our method has potential for broad applications.
> > >
> > > 2. Regarding the analysis of internal recognition mechanisms:
> > > To further clarify the meaning of our analysis: The differences in model accuracy when facing different types of queries directly reflect its internal ability to differentiate and recognize these queries. Specifically:
> > >
> > > - Contextual conflict queries: The model performs well, indicating its internal mechanism can effectively identify such conflicts.
> > > - Parametric knowledge conflict queries: Lower accuracy suggests the model's internal mechanism struggles to differentiate these queries from non-conflict queries.
> > >
> > > This differentiation explains why the model exhibits varying refusal capabilities across different types of conflict queries.
> > >
> > > While we acknowledge there is room for improvement in our current work, we believe that under the existing experimental design and analytical framework, this research provides valuable insights and viable improvement methods.

---

### Official Review · Reviewer_Dn9U · 2024-11-04

**Soundness:** 3
**Presentation:** 3
**Contribution:** 2
**Rating:** 6
**Confidence:** 4

**Summary:**

This paper proposes a benchmark to evaluate LLM's ability for role playing from the aspect of whether these LLMs can reject conflicting queries. It then uses linear probing and t-SNE to analyze why different models behave differently.

**Strengths:**

1. propose a well-motivated benchmark
2. the data construction pipeline is plausible
3. conduct interpretability experiment to analyze results
4. develop a model editing method based on the representation discoveries
In general, this paper raises interesting research questions and also conduct in-depth analysis.

**Weaknesses:**

1. The editing method does not analyze how much the method affects other non-relevant questions, such as questions independent to the role-playing. So the general accuracy of the thresholding method needs a comprehensive analysis

**Questions:**

1. Figure 2 shows that the model is bad at parametric conflicting queries. But why is the accuracy high at early layers and then decreases as layer number increase? And also why for non-conflicting queries, the accuracy starts low at early layers?

---

> ### Author Response · Authors · 2024-11-19
> **Response to W1: General Performance**
>
> As we mentioned in Section 6.3.1 of our paper, to thoroughly evaluate whether our proposed editing method enhances the refusal capabilities of RPAs while not affecting their ability to handle non-role-playing-related questions, we employed MT-Bench as a comprehensive evaluation benchmark. MT-Bench is a multidimensional evaluation tool designed to assess the performance of large language models across various tasks and scenarios.
>
> We conducted detailed evaluations across eight dimensions of MT-Bench, and the results are presented in Tables 1 and 2:
>
> **Table 1**: Performance of Llama-3-8B-Instruct under different methods
>
> | Method                   | Writing | Roleplay | Reasoning | Math  | Coding | Extraction | STEM  | Humanities | Average |
> |--------------------------|---------|----------|-----------|-------|--------|------------|-------|------------|---------|
> | **FT**                   | 8.06    | 7.05     | 4.40      | **6.15** | 5.60   | 7.55       | **9.16** | 9.40       | 7.16    |
> | **LoRA**                 | 8.15    | 7.70     | **6.55**  | 4.70  | **5.75** | 7.20       | **9.16** | **9.80**   | **7.37** |
> | **Representation Editing** | **8.98** | **8.30** | 4.35      | 5.35  | 5.35   | **7.78**   | 9.05  | **9.80**   | 7.36    |
>
> **Table 2**: Performance of Llama-3.1-8B-Instruct under different methods
>
> | Method                   | Writing | Roleplay | Reasoning | Math  | Coding | Extraction | STEM  | Humanities | Average |
> |--------------------------|---------|----------|-----------|-------|--------|------------|-------|------------|---------|
> | **FT**                   | 7.47    | 7.55     | 3.80      | 6.40  | **6.50** | 6.53       | 7.65  | 9.20       | 6.88    |
> | **LoRA**                 | 9.10    | 8.00     | 5.30      | **6.55** | 6.30   | 7.00       | 8.65  | **10.0**   | 7.61    |
> | **Representation Editing** | **9.15** | **8.15** | **5.85**  | 6.25  | 5.95   | **7.30**   | **9.70** | 9.93       | **7.78** |
>
> Based on the results in Tables 1 and 2, we can clearly see that the Representation Editing method enhances the models' performance in the Roleplay dimension compared to both LoRA and FT methods. Simultaneously, the impact of the Representation Editing method on other dimensions is minimal, and we even observe slight performance improvements in certain areas. This demonstrates that our method **successfully improves the models' role-playing abilities while maintaining their general capabilities**. These results fully attest to the effectiveness of the Representation Editing method.

---

> ### Author Response · Authors · 2024-11-19
> **Response to Q1: Changes in Probe Accuracy**
>
> **1. Input Processing in Probe Experiments:**
>
> In our probe experiments, the input to the probe is the representation of the last token of the query. To avoid semantic inconsistencies that may arise from different tokens, we standardized the last token of all queries to be the same character (the end-of-text token, for example, `<|eot_id|>` in the Llama model). This approach ensures that the inputs received by the probe are semantically consistent, reducing interference caused by semantic differences among tokens.
>
> **2. Reason for High Accuracy in Early Layers:**
>
> The shallow layers of Transformer models (those close to the input layer) primarily handle and encode the basic semantic and syntactic information of input tokens. In this scenario, since the last token of all queries is standardized to the end-of-text token, the representation vectors in the shallow layers are very similar semantically. This high similarity makes it challenging for the probe to distinguish between different types of queries in the early layers. The probe model tends to make the same prediction for most samples (e.g., predicting 0 or 1 for all). As we mentioned in Appendix C.2. For each type, we use the same amount of data, but since there are four types of conflicting queries and only one type of non-conflicting queries, conflicting queries are the majority. Therefore, the probe is inclined to predict conflicting queries, leading to higher accuracy for conflicting queries and lower accuracy for non-conflicting queries in early layers.
>
> **3. Reason for Changes in Accuracy as Layer Number Increases:**
>
> As the layers deepen, the model gradually integrates higher-level semantic and contextual information. The representations in the deeper layers contain more complex features related to the query background and context. These features cause different types of queries to become more dispersed and diverse in the representation space. Consequently, the features contained in the representations increase accordingly, leading to changes in the probe's accuracy.
>
> Our observations are consistent with existing research findings. For example, in their studies, [1] and [2] point out that the shallow layers of the model are mainly responsible for encoding basic grammatical and semantic information, while the deeper layers integrate more complex contextual and semantic relationships. This explains why the probe exhibits higher/lower accuracy for conflicting/non-conflicting queries in the shallow layers and why the accuracy varies in the deeper layers.
>
> [1] Zou, Andy, et al. "Representation Engineering: A Top-Down Approach to AI Transparency." *arXiv preprint* arXiv:2310.01405 (2023).
>
> [2] Liu, Wenhao, et al. "Aligning Large Language Models with Human Preferences through Representation Engineering." *arXiv preprint* arXiv:2312.15997 (2023).

---

> ### Comment · Reviewer_Dn9U · 2024-11-26
> **Thank you for your detailed reply**
>
> Thank you the detailed the reply. I will keep my weak accept score

---

### Official Review · Reviewer_oci5 · 2024-11-05

**Soundness:** 3
**Presentation:** 3
**Contribution:** 3
**Rating:** 6
**Confidence:** 4

**Summary:**

The paper investigates the challenges faced by Role-Playing Agents (RPAs) in handling conflicting queries that contradict their role-play knowledge. The authors perform in-depth analysis of RPAs' performance across different types of requests, including contextual and parametric knowledge conflicts. They identify the presence of "rejection regions" and "direct response regions" in representation space. Based on which, they propose a lightweight representation editing method that enhances the refusal accuracy of RPAs while maintaining their overall role-playing capabilities.

**Strengths:**

1.The paper studies an interesting and important problem. Enhancing RPA’s ability to refuse questions they do not know could important implications for various applications, like virtual assistants and game design.

2.I like the representation analysis part. I believe it is a novel finding to identify "rejection regions" and "direct response regions". The analysis provides adequate motivations for the proposed representation editing method.

3. The authors provide extensive experiments to demonstrate the effectiveness of the proposed solutions.

**Weaknesses:**

1. The paper could benefit from including user-centric studies to evaluate the real-world impact of enhanced refusal capabilities.

2.While the empirical findings are strong, the theoretical underpinning of the rejection and response regions may require further exploration to enhance understanding.

**Questions:**

Please refer to the Weaknesses above.

---

> ### Author Response · Authors · 2024-11-19
> **Response to W1: benefit from including user-centric studies**
>
> Thank for your suggestion. As `Reviewer tFS5` mentioned, *"It seems from Figure 1 that RPAs mainly converse with users."* Our research is indeed user-centric, which is reflected in the following aspects:
>
> - **Design of User Interaction Scenarios:** When constructing the RoleRef evaluation benchmark, we designed various query scenarios that involve user interactions, including role setting conflicts and knowledge conflicts. The aim is to simulate real-world issues that users might encounter when using RPAs, ensuring our study addresses practical user concerns.
>
> - **Consideration of Avoiding Over-Refusal:** While evaluating RPAs' refusal capabilities, we also emphasize preventing the model from over-refusing. We ensure that users can still receive effective responses within a reasonable scope, thereby enhancing the overall user experience and satisfaction.
>
> - **Evaluation of Role-Playing Effectiveness:** We placed special focus on role-playing consistency and the model's ability to recognize knowledge boundaries. These capabilities directly impact user immersion and trust during interactions with RPAs.
>
> - **Practical Application Examples:** In our introduction, we discussed the applications of RPAs in video games, virtual assistants, and educational tools. These are real-world, user-facing scenarios that highlight the user-oriented nature of our research.
>
> By centering our study around user interactions and experiences, we believe our work effectively evaluates the real-world impact of enhanced refusal capabilities in RPAs.

---

> ### Author Response · Authors · 2024-11-19
> **Response to W2:  Theoretical Underpinning Of The Rejection and Response Regions**
>
> Delving deeper into the theoretical underpinnings of the **direct rejection regions** and **response regions** indeed helps enhance our understanding of model behavior.
>
> Regarding your suggestion to further explore the theoretical foundations of the rejection regions and response regions, we highly value it and have provided preliminary theoretical definitions of these regions in the revised manuscript.
>
> Specifically, given the representation vector $\\mathbf{h}^l$ at the $l$-th layer of the model and the rejection direction vector $\\mathbf{d}'^l$, we determine the position of the input query in the representation space by calculating the similarity between them, thereby deciding the response strategy the model should adopt. The cosine similarity is defined as:
>
> $$
> \\text{sim}(\\mathbf{h}^l, \\mathbf{d}'^l) = \\frac{\\mathbf{h}^l \\cdot \\mathbf{d}'^l}{\\|\\mathbf{h}^l\\| \\|\\mathbf{d}'^l\\|}
> $$
>
> Based on the similarity, we can define the rejection region and the response region:
>
> - **Direct Rejection Region**: When $\\text{sim}(\\mathbf{h}^l, \\mathbf{d}'^l) \\geq \\theta$, where $\\theta$ is a preset threshold, indicating that the input is highly related to the rejection direction, and the model should tend to refuse to answer.
>
> - **Response Region**: When $\\text{sim}(\\mathbf{h}^l, \\mathbf{d}'^l) < \\theta$, indicating that the input does not belong to a conflicting query, and the model should proceed to generate a direct response.
>
> Mathematically expressed as:
>
> $$
> \\text{Decision} =
> \\begin{cases}
> \\text{Refusal}, & \\text{if } \\text{sim}(\\mathbf{h}^l, \\mathbf{d}'^l) \\geq \\theta \\\\
> \\text{Direct Response}, & \\text{if } \\text{sim}(\\mathbf{h}^l, \\mathbf{d}'^l) < \\theta
> \\end{cases}
> $$
>
> Through the above definitions, we preliminarily delineate the positional relationships of the rejection region and the response region in the representation space. This theoretical framework provides an initial basis for understanding the model's decision-making process, helping to explain why the representation editing method can effectively guide conflicting queries toward the rejection region.
>
> We also recognize that, due to the high-dimensional and nonlinear characteristics of large language models, conducting an in-depth theoretical analysis poses certain challenges. However, our method has demonstrated robust and significant performance improvements across multiple models and scenarios, validating its practicality and effectiveness.

---

> > ### Comment · Reviewer_oci5 · 2024-11-26
> >
> > Thank you for your reply. It addresses some of my concerns. I think the research problem is interesting and may have broad applications. I will keep my Weak Accept rating.

---

### Official Review · Reviewer_tFS5 · 2024-11-06

**Soundness:** 3
**Presentation:** 4
**Contribution:** 3
**Rating:** 6
**Confidence:** 3

**Summary:**

The paper targets understanding the limitation of role-playing agents (RPAs) in recognizing and responding to hard queries that conflict with their knowledge. To this end, the authors develop an evaluation benchmark including conflicting and non-conflicting requests to assess RPAs' ability to identify conflicts and refuse to answer in hard cases without over refusing, thereby derive some findings about the RPAs' performance against different conflicts as well as the underlying reasons through a representation-level analysis. They also proposed the editing approach to let RPAs refuse requests concerning conflicts.
The significance of the work is without a doubt: RPAs are not supposed to answer any question which they do not have to answer to, instead of always trying the best to give an answer.

**Strengths:**

The refusal capabilities appear to be an important issue indeed. It is interesting to see work towards this direction to improve the RPAs' performance.

The paper is well written -- the study follows a step by step procedure, from evaluation design to comparison, and finally, to methods to improve the RPAs. It is well organized and easy to follow, with lots of examples to ease understanding.

Given the examples provided in the paper, it is convincing how the work introduced could help on the RPAs, delivering tangible understnading of the impact within the datasets/scenarios used.

**Weaknesses:**

The research methodology is kind of straightforward; it is simple what the authors intend to do and they made it via a sound process. For the same reason, it is not obvious what the challenges are for this study.

The representation editing method intervenes with the representations generated by the model to enhance the refusal ability for conflicting cases. It is compared with several fine-tuning methods designed for LLMs. But essentially, it may not be of the same nature as the compared methods. It is worth to explore how the proposed method works with other methods than LoRA.

There might be some more introduction to give readers a better understand of what RPAs are and what they do in scenarios like video games, etc. It seems from Figure 1 that RPAs mainly converse with users. If the focus is on queries and responses, this should be made clear to narrow the study of RPAs' capabilities to query-response.

It might be necessary to also consider the computation overhead in the proposed design.

**Questions:**

Are there any other studies around the refusal capabilities of RPAs? If so, they should definitely be included in the paper and discussed in comparison with the work presented in the paper.

---

> ### Author Response · Authors · 2024-11-19
> **Response to W1: Challenges in Our Work**
>
> Although our research methodology looks like straightforward, there are many significant challenges in enhancing the model's refusal capabilities.
>
> 1. **Enhancing the model's refusal ability is a challenging task.** Even current state-of-the-art models (such as GPT-4o) cannot fully and correctly recognize and refuse queries that conflict with their role knowledge. In our [Failure Cases](https://anonymous.4open.science/r/Failure-Cases-of-Tell-Me-What-You-Don-t-Know-A3B4/Failure%20Cases.csv), we provide specific examples that demonstrate the challenges in this area of research.
>
> 2. Existing related work mainly **focuses on specific types of conflicts**, such as time hallucinations, with little research on other types of conflict scenarios. Moreover, previous **evaluations mainly concentrate on certain aspects of role-playing** (such as personality consistency), lacking a systematic evaluation of the model's refusal capabilities.
>
> 3. Regarding the performance differences for different types of queries, we **conducted an in-depth interpretability analysis**, revealing the internal mechanisms of the model under various conflict scenarios. Based on the analysis, we **proposed a representation editing method** that can enhance the model's refusal capabilities without additional training. This work not only provides new perspectives for understanding and improving the model but also introduces methodological innovations.
>
> In summary, our research not only overcomes the challenges of enhancing the model's refusal capabilities but also makes corresponding contributions in terms of scenarios, evaluation dimensions, methodologies, and analysis. We expanded the research on conflict scenarios, provided a systematic evaluation of refusal capabilities, introduced new methods to improve model performance, and conducted an in-depth analysis of the model's internal working mechanisms.

---

> ### Author Response · Authors · 2024-11-19
> **Response to W2: Choice of Baseline**
>
> Firstly, we chose to compare the representation editing method with fine-tuning methods based on their **shared goal**: enhancing the model's refusal capabilities while preserving its original performance as much as possible. Many previous studies (such as references [1][2][3]) have employed fine-tuning or LoRA to improve the model's refusal abilities. Therefore, comparing our method with these commonly used and effective approaches is practical and valuable.
>
> Our method is similar to LoRA in that both introduce a bias term into the model's representations to influence the model's output responses. However, the key differences are:
>
> 1. **No Training Required**: Our representation editing method does not require any additional training or fine-tuning of the model. Instead, it directly intervenes in the model's internal representations during inference. This makes our method more lightweight and efficient, suitable for scenarios with limited computational resources.
>
> 2. **Different Implementation Approaches**: The LoRA method adjusts model parameters by adding trainable low-rank adapter to the model, necessitating additional training steps. In contrast, our method directly adds a bias to the model's representations without changing the model's parameters or structure.
>
> Experimental results have demonstrated that our representation editing method can effectively enhance the model's refusal capabilities, achieving similar or even better results compared to LoRA without affecting the model's original performance.
>
> [1] Brahman, Faeze, et al. "The art of saying no: Contextual noncompliance in language models, 2024." URL https://arxiv. org/abs/2407 12043.
>
> [2] Chen, Lida, et al. "Teaching Large Language Models to Express Knowledge Boundary from Their Own Signals." arXiv preprint arXiv:2406.10881 (2024).
>
> [3] Cheng, Qinyuan, et al. "Can AI Assistants Know What They Don't Know?." arXiv preprint arXiv:2401.13275 (2024).

---

> ### Author Response · Authors · 2024-11-19
> **Response to W3: Details of RPA Application Scenarios and Why They Are Limited to Query-Response Scenarios**
>
> 1. **On the Importance of RPAs:**
>
>    RPAs play a significant role in the field of artificial intelligence, especially in interactive applications such as video games, virtual assistants, and education. In video games, RPAs are used as Non-Player Characters (NPCs) to create realistic and personalized characters, making the game environment more vivid and immersive. As NPC dialogue systems, they can generate dynamic and context-appropriate responses based on player input [4]. This makes interactions with NPCs more engaging and lifelike, reduces repetitive dialogue, and provides a more exploratory experience within the game. Additionally, RPAs are applied in areas like virtual customer service and online education, acting in specific roles to provide personalized services to users. Therefore, a deep understanding and enhancement of RPAs' capabilities are crucial for improving the quality and efficiency of human-computer interaction.
>
> 2. **On the Importance of Refusal Abilities:**
>
>    In interactions with users, RPAs may encounter questions that are beyond their knowledge scope, violate their role settings, or involve sensitive information. If RPAs lack appropriate refusal abilities, it can lead to the following problems:
>
>    - **Providing incorrect or misleading information:** This may cause confusion for users or even bring about safety and ethical risks.
>    - **Breaking role consistency:** If RPAs answer questions that are inconsistent with their character, it may reduce user immersion and trust. For example, in a game, if an RPA portraying a medieval knight is asked about modern technology and cannot appropriately refuse or respond, it might break the game's authenticity and affect the player's experience.
>
>    Therefore, enhancing RPAs' refusal abilities is vital for building reliable, professional, and consistent human-computer interaction systems.
>
> 3. **Why the Research Focuses on Query-Response:**
>
>    Our research concentrates on the query-response interactions of RPAs mainly because:
>
>    - **Core interaction method:** In most applications, the primary form of interaction between RPAs and users is through dialogue—exchanging information and completing tasks via queries and responses.
>    - **Focusing on key issues:** By studying query-response interactions, we can more directly analyze RPAs' ability to recognize and handle different types of requests (including conflicting requests), gaining deeper insights into the models' behavioral mechanisms.
>
> 4. **Generalization of the Research:**
>
>    Although our experiments are conducted within query-response scenarios, we believe that the designed scenarios, evaluation methods, and strategies for enhancing refusal abilities have broad applicability:
>    - **Universality of scenario design:** Our conflict scenarios are based on the two main knowledge sources of RPAs—contextual knowledge and parametric knowledge. These types of conflicts are common across various RPA applications and are not limited to specific domains.
>    - **Generality of evaluation methods:** The evaluation metrics and methods we propose can be used to assess RPAs' refusal abilities and role-playing capabilities in different applications, offering wide-ranging reference value.
>    - **Extensibility of enhancement methods:** Our improvement strategy based on the representation editing approach has already proven effective in query-response scenarios across different models. We believe it can also be extended to more complex interaction scenarios, helping to improve RPAs' performance in various applications.
>
> [4] Gallotta, R., et al. "Large Language Models and Games: A Survey and Roadmap. arXiv 2024." arXiv preprint arXiv:2402.18659.

---

> ### Author Response · Authors · 2024-11-19
> **Response to W4: Computation Overhead**
>
> The representation editing method does not incur significant additional computational overhead. We analyze the computational overhead of our method mainly from two aspects: training overhead and inference overhead.
>
> **1. Training Overhead:**
>
> As we have shown in Table 4 of our paper, our method does not involve any trainable parameters. Specifically, we only need to precompute and store the rejection vectors, which can then be simply added to the model's internal representations during practical applications. Therefore, compared to Fine-Tuning (FT) and LoRA, the computational overhead during the training phase of the representation editing method is nearly zero.
>
> **2. Inference Overhead:**
>
> During inference, our method only requires a simple vector addition operation between the precomputed rejection vectors and the current internal representations of the model. This operation has a computational complexity similar to the adapter modules in LoRA. Since this operation is extremely lightweight, its impact on inference time and computational resources is almost negligible. Therefore, our method does not introduce significant additional overhead during the inference phase either.

---

> ### Author Response · Authors · 2024-11-19
> **Response to Q1:  Any Other Studies Around The Refusal Capabilities Of RPAs**
>
> As we mentioned in the Related Work section, research on the refusal capabilities of models mostly focuses on general question-answering scenarios. The goal is to enable models to recognize their own knowledge blind spots when faced with queries that conflict with their parametric knowledge and appropriately refuse to answer, to avoid providing incorrect or misleading information.
>
> In the field of Role-Playing Agents (RPAs), the studies most relevant to our work include [5] and [6]:
> - **[5]** In their research, they focus on the performance of role-playing chatbots regarding temporal knowledge consistency. They explore the issue of hallucinations when models handle temporal information and evaluate temporal consistency in role-playing.
> - **[6]** This work primarily addresses the problem of temporal hallucinations in RPAs and proposes mitigation strategies to make models more accurate when dealing with queries involving temporal information.
>
> Although the above studies involve the refusal capabilities of RPAs to some extent, they mainly focus on handling specific temporal hallucinations. Our work aims to conduct a systematic and comprehensive study of the refusal capabilities of RPAs. Specifically:
> 1. **Diverse Conflict Scenarios**: We have designed multiple conflict scenarios, including conflicts with the role's contextual knowledge (such as role setting conflicts and role description conflicts) and conflicts with the role's parametric knowledge (such as factual knowledge conflicts and absent knowledge conflicts). These scenarios encompass various aspects of the role's knowledge, not limited to temporal information.
> 2. **In-depth Analysis of Refusal Capabilities**: We not only evaluate the models' refusal capabilities in different conflict scenarios but also analyze the differences in the models' internal representations when handling these conflicts using techniques like linear probes and t-SNE visualization. We explore why models perform differently on different types of conflicting queries.
> 3. **Proposed Improvement Method**: Based on our analysis, we propose a representation editing method aimed at enhancing the models' refusal capabilities without compromising their overall role-playing performance. This method differs from traditional fine-tuning or training strategies and is lightweight and efficient.
>
> Therefore, the difference between our work and existing research lies in our more comprehensive and in-depth exploration of the refusal capabilities of RPAs, covering more types of conflicts, and proposing a new improvement method that fills a gap in current research.
>
> [5] Ahn, Jaewoo, et al. "TimeChara: Evaluating Point-in-Time Character Hallucination of Role-Playing Large Language Models." *arXiv preprint* arXiv:2405.18027 (2024).
>
> [6] Sadeq, Nafis, et al. "Mitigating Hallucination in Fictional Character Role-Play." *arXiv preprint* arXiv:2406.17260 (2024).

---

### Author Response · Authors · 2024-11-19
**Global Response**

We express our gratitude to all the reviewers for their valuable insights and constructive feedback! We are pleased to hear that you appreciated our contributions to enhancing the refusal capabilities of RPAs through representation space analysis and editing. We would like to highlight some of the strengths of our work as noted by the reviewers:

1. **Interesting and Important Problem** (Reviewer `tFS5`, `oci5`, `Dn9U`, `j72b`, `GGuM`)

2. **In-depth Analysis**: Reviewers (Reviewer `oci5`, `Dn9U`)

3. **Comprehensive Experiments** (Reviewer `oci5`, `Dn9U`).

4. **Well-Written and Easy to Understand**(Reviewer `tFS5`).

A major concern shared by several reviewers was the need for more detailed explanations regarding the selection of baselines. To address this, we have clarified our rationale for choosing Fine-tuning and LoRA as baseline methods. These methods are widely used and recognized for enhancing model capabilities, making them suitable benchmarks for evaluating the effectiveness of our representation editing approach. By comparing our method against these established techniques, we aim to provide a fair and comprehensive assessment of its performance.

Another concern raised was the potential bias introduced by using GPT-4 for data generation and evaluation. We have clarified that our choice was based on GPT-4's high alignment with human evaluations, as supported by existing research. Furthermore, we incorporated human evaluation steps to ensure data quality and diversity, minimizing any potential biases.

Finally, we express our sincere gratitude to the reviewers for recognizing the contributions of our proposed method to enhancing the refusal capabilities of RPAs. Our work offers a new perspective on improving model safety and reliability through representation editing, providing a lightweight and efficient alternative to more resource-intensive methods. We hope this study inspires further research in developing robust and trustworthy AI systems.

---

### Meta-Review · Area_Chair_ETJi · 2024-12-16

**Metareview:**

This paper received a split decision from the reviewers, three were marginally in favor of acceptance and one was marginally for rejection and one was certain about rejection.  In my reading of the paper, I also agree that there are many valuable insights presented, but that the evaluation is insufficiently strong.  The authors propose a representation editing approach that enables a model to refuse to answer (without retraining the model) and they show that this generates improvement under two types of evaluation.  The results show marginal improvement for the experiments that the did, (See Table 4) indicating that they have good insights, but the impact is not unambiguous.  Although rejection is never the desired outcome, at present this seems like a weaker paper that could be made stronger with revision.

The authors themselves sum up the critiques that were given during the review process:

A major concern shared by several reviewers was the need for more detailed explanations regarding the selection of baselines. To address this, we have clarified our rationale for choosing Fine-tuning and LoRA as baseline methods. These methods are widely used and recognized for enhancing model capabilities, making them suitable benchmarks for evaluating the effectiveness of our representation editing approach. By comparing our method against these established techniques, we aim to provide a fair and comprehensive assessment of its performance.

The clarification was considered insufficient and more evaluation is requested.

Another concern raised was the potential bias introduced by using GPT-4 for data generation and evaluation. We have clarified that our choice was based on GPT-4's high alignment with human evaluations, as supported by existing research. Furthermore, we incorporated human evaluation steps to ensure data quality and diversity, minimizing any potential biases.

The LLM bias problem is real and more prevalent in more capable models.  It would be great if you included some form of human evaluation and described it in the paper. You use "human evaluation steps" in your argument against the criticism of LLM bias but the only evidence that I see of the human evaluation aspect is your comment that you used human spot checking in your data construction process:

Additionally, in our dataset construction process, we did not solely rely on model outputs; we also incorporated a human evaluation step. Specifically, after data generation, we randomly selected a portion of the samples for manual verification and assessment to ensure the quality and diversity of the data. Through this approach, we aim to minimize the impact of any biases that might arise from model self-evaluation.

Please include a more detailed description of your human evaluation process if you believe that it counters the LLM bias issue.

Overall, it possible in a future submission please hit on key points of evaluation, including dataset  instruction, clearly in the main part of the paper.

**Additional Comments On Reviewer Discussion:**

During the author rebuttal period, only two reviewers commented on the rebuttals and these were the two who gave it a negative rating.  Neither of them were swayed by the author's rebuttals and in fact the slightly negative reviewer (5) commented that they agreed with the comment of the highly negative reviewer (3) but just not to the same extent.

Overall, my feeling is that the two slightly positive saw the value of the insights of the paper.  I did too.  I think it has potential, it is just insufficiently rigorous and the writing could be improved.  I think the slightly positive reviewers were not against it but not particularly enthusiastic about it either as evidenced by their failure to return to the discussion after their initial review despite reminders.

---

### Decision · Program_Chairs · 2025-01-22

Reject